# Reliability Analysis during the Life Cycle of a Technical System and the Monitoring of Reliability Properties

**Alena Breznická, Marcel Kohutiar *** , **Michal Krbaťa** , **Maroš Eckert and Pavol Mikuš**

Faculty of Special Technology, Alexander Dubcek University of Trenčín, 911 06 Trenčín, Slovakia;
alena.breznicka@tnuni.sk (A.B.); michal.krbata@tnuni.sk (M.K.); maros.eckert@tnuni.sk (M.E.);
pavol.mikus@tnuni.sk (P.M.)
* Correspondence: marcel.kohutiar@tnuni.sk

**Abstract:** The present review deals with the issues of ensuring and maintaining high reliability during the entire life cycle of a technical system in the engineering sector, i.e., determining the requirements, design, research, development, production, installation, operation, and disposal of the product. Owing to their reliability, special attention is given to the technical systems in companies for several reasons. These mainly include the need to achieve reliability and safety by using the product, but also in terms of economic, social, and ecological aspects. We understand reliability as a primary feature, and during the design of new products, it is necessary to predict its course and characteristics appropriately. Reliability needs to be analyzed with the help of partial reliability properties throughout the entire life cycle of the product. To achieve the required level of reliability, we use reliability analyses, which are successfully implemented to examine and predict reliability indicators. The present review provides a comprehensive overview of reliability analysis and offers a mathematical basis for individual sub-indicators. In the article, the analysis of system reliability is described as a process, the essence of which is the acquisition, examination, and organization of specific information. This information is significant for the given system and necessary for making decisions according to the specified goals, which ensure an objective assessment of the overall level of reliability.

**Keywords:** reliability; life cycle; analysis; methods; product





## 1. Introduction

The issues of ensuring and maintaining high reliability throughout the lifetime of a technical system, i.e., determining the requirements, design, research, development, production, installation, operation, and disposal of the product, are given special attention in companies for several reasons. Alessandro Birolini [1] argues that these are mainly individual and social needs, including the need to fulfil predetermined tasks under any conditions; economic needs, which express the effort to use technology at optimal costs; ecological needs, which help to preserve the environment; and hygiene and safety elements, which ensure the protection of human and employee health during operation but are also important in the event of a malfunction or crash. The assessed technical systems usually consist of different components that can be considered as a tree structure according to their relationship with the content [2]. An important element in ensuring reliability is the management and leadership approach of the company that is responsible for the production of the technical system. Currently, elements of the lean approach are actively used [3].

The reliability of objects can be understood as their ability to fulfil the required functions within the specified time while observing the operational parameters given by the technical conditions [4]. Reliability management is a coordinated activity of the manufacturer and the user through which measures are planned and implemented, determining the requirements, design, research, development, construction, production, testing, operation,

and disposal of the product in order to ensure the reliability required by the user. The reliability management system includes research, development, production, and operational control, as well as executive components and stages [5]. In product development, important specification and design decisions must be made at various stages of the life cycle, which include the design, manufacturing, operations, and support stages. However, making these decisions becomes more complex when a multi-disciplinary team of stakeholders is involved in system-level or subsystem-level architecture and design decisions [6,7].

To achieve the required level of reliability, we use reliability analyses. These reliability analyses are used to assess a system's reliability. To achieve the desired level of reliability, separate steps involving design modification, iteration, and optimization are undertaken. Different qualitative and quantitative methods are used, employing mathematical modeling and appropriate simulations [8]. Reliability analyses are carried out at the stages of conceptualization and requirement setting, design and development, production, use, and disposal. They are also applicable at the stages of operation and maintenance, primarily for the evaluation and determination of reliability indicators and to assess the fulfilment of the specified requirements. This means that such analyses are relevant at all stages of a product's life cycle. System reliability analysis is a process whose essence is the acquisition, examination, and arrangement of significant information for the given system and necessary for making decisions according to the specified goals [9,10]. Research usually takes place using system models. The final product of this process is a set of information concerning the properties of the system model [11]. The model can be modified during the analysis. The analysis must be performed according to clearly defined rules and procedures, so that the analysis process is repeatable and always leads to the same results [12,13].

Reliability analyses are also used to predict indicators of system reliability/freedom from failure, readiness, sustainability, and security [14].

The tasks of the reliability management system include the following:

- Investigating and revealing the factors effecting the reliability of products in operation and their qualification;
- The application and development of a reliability theory;
- The determination and implementation of methodologies, and the evaluation of the laboratory, operational, and other tests;
- The monitoring, collection, and processing of user-based operational reliability data;
- The application of knowledge acquired in the determination of product requirements and in the design, research, development, production, and operation stages for the preparation and training of executives and managers;
- The coordination of activities between components of the reliability management system [15,16].

## 2. Content and Tasks of the General Reliability Analysis Procedure

### 2.1. System Definition and Requirements

The first step is to define the analyzed system, types of operation, and the functional relationships to higher levels and neighboring systems or processes [17]. A list of all requirements for service life, the absence of faults, maintainability and maintenance assurance, readiness, safety, storability, and diagnosability is drawn up [18]. The definition includes system characteristics and properties related to these requirements, including environmental conditions, operation, and maintenance requirements. For control systems linked to controlled objects, e.g., production and technological systems or complex technical equipment, we define the safety and protection functions resulting from safety requirements [19].

### 2.2. Definition of Failure States, Critical Failure States, and Limit State

They are defined, refined, and supplemented through the analysis of system failure states (failure states critical for safety), their criteria and conditions based on the functional requirements of the system, the expected operation, and the operating environment [20].

Among these activities are analyses of the consequences of failures (differentiation of failures or failure states into the categories of critical, essential, and non-essential). We can use various methods of qualitative and quantitative technical system reliability analysis [21].

For complex systems, it is necessary to distinguish the following states:

- A permanent failure state, as a result of a sudden and complete failure or even a degradation failure, i.e., a failure condition that lasts until maintenance is performed after the failure (so-called permanent disruption of the ability's operation, or hard failure) [22];
- An occasional (transient) failure state (so-called failure, short-term disruption of operability, or soft failure) that lasts for a certain period, after which the object regains the ability to perform the required function, without performing any maintenance activity after the failure. It is a consequence of an occasional (transient) disorder [23,24]. Occasional malfunctions require the elimination of their consequences via appropriate measures.

### 2.3. Distribution of Requirements for Subsystems, Blocks, and Components

If a mathematical analysis is required, it is recommended to divide the preliminary design of the system requirements into requirements for its subsystems, blocks, elements, etc. To do this, we must divide the system into lower system parts, elements, while respecting the point of view that the elements form sub-integrated structural or functional parts, which are at least one degree simpler than the system and are supplied by subcontractors or subject to renewal as a whole [25,26].

### 2.4. Reliability Analyses

Custom analyses of partial properties, e.g., reliability, safety, objects, or systems generally have two overlapping approaches, which can be characterized as qualitative and quantitative [27]. Quantitative analyses are always preceded by a qualitative part of the analysis. Successful analysis requires teamwork, and its efficiency is significantly increased by using appropriate software.

A similar approach is also applied to the choice of system failure analysis methods, referred to as risk analysis methods [28,29]. Basically, the same methods are used, sometimes slightly modified, but more emphasis is placed on managerial assurance and responsibility for performing analyses, especially for when deciding on measures aimed at correcting the causes and consequences of malfunctions.

### 2.5. Assessment of Compliance with Requirements and Corrective Measures

In the last step, the results are evaluated and compared with the specified requirements, and at the same time, additional activities are carried out. For example, a review of the system design, the determination of "weak" places/critical or high-risk types of failure states and units, and fail-safe options and mechanisms are considered, etc. Alternative reliability improvement procedures (backup requirement scheduling, parameter tracking, fault condition detection, system reconfiguration procedures, maintainability, component interchangeability, and repair procedures) are being developed [30].

### 2.6. Qualitative and Quantitative Analyses of Reliability

We generally recognize two types of reliability analysis methods:

Qualitative analyses that allow the state of the analyzed object/system to be expressed using verbal expression (e.g., keywords, syntagms, or short sentences) and/or the value of a parameter expressed using a logical function or representation (e.g., in the form of assignment to individual classes) [31].

The parameters of qualitative methods of reliability analysis have the characteristics of attributive signs and acquire positive integer values.

Qualitative analyses include the study and investigation of types, causes, dependencies, manifestations, and consequences of disorders, and the possibility of preventing them

or eliminating their causes or consequences. In the case of using qualitative analyses in the pre-production stages, analyses of potentially possible types of failures, their causes, dependencies, and consequences are carried out [32]. They generally contain an analysis of the functional structure of the system, the determination of the system failure state types, components, failure mechanisms, manifestations, and consequences of failures, the consideration of the maintainability of units, the compilation of safety models and fault-free operation, the determination of possible strategies for maintenance (prevention of failures) and repairs, etc. Part of the analysis is the determination and description of the required functions of the system, the conditions of its use, the appropriate choice of the system function representation in terms of reliability (e.g., functional block diagrams, etc.), the construction of the system reliability model, i.e., graphical representation, and a mathematical or other description of the system structure, expressing the relationship between this structure and the properties of the system with a link to its reliability [33].

Quantitative analyses enabling the state of the analyzed object/system to be expressed using the parameter value of the expressed reliability indicator and possibly other statistical quantities. The parameters of quantitative methods of reliability analysis have the characteristics of variable signs and are real numbers [34].

Quantitative properties of reliability are numerically expressed using indicators, i.e., the distribution of random variables. They enable features such as indicators of service life, readiness, and other partial features of the system to be determined depending on the reliability on the indicators of fault-freeness, maintainability and maintenance assurance, stress conditions, or the loading of the elements, and they allow a suitable alternative design of the system to reach a pre-specified level. In the pre-production stages, when the system does not physically exist, they have the characteristics of predictions. Their typical feature is to interactively determine their value with gradual refinement, linked to the sophistication of the design with the transition to the final structure of the projected system. In quantitative analyses, data are obtained or determined for mathematical models.

Predictions of system indicators are generally the result of a more complex multi-step process. Based on careful consideration of the purpose and significance of individual steps of the general analysis procedure, the evaluation of all possibilities, advantages and disadvantages of individual methods, and availability of necessary data and information, a suitable method for analysis is selected. Most of the time, the individual methods complement each other because, as a rule, no single analysis method is exhaustive enough to handle all possible aspects required for the evaluation of a particular system [35]. In order to assess the results, but also the principles and means by which they were obtained, it is necessary to provide a fairly extensive set of additional information to the actual numerical values, the recommended scope of which includes, e.g., standard IEC 60863: 1992 (01 0621) (IEC—International Electrotechnical Commission) [36].

### 2.7. Inductive and Deductive Analyses

When analyzing reliability through inductive or deductive analysis, there are two different methodological procedures for system reliability analysis: inductive, i.e., a "bottom-up" procedure, and deductive, i.e., a "top-down" procedure. The inductive procedure is based on the progression of the analysis from specific elementary problems to more general and global problems. From the analysis of functions and malfunctions of elements (including their combinations) to the lowest level of system breakdown, one proceeds to the analysis of malfunctions and their consequences on superior systems up to malfunctions of the entire system. This procedure is applied, for example, in the failure mode and effects analysis (FMEA) method, where the consequences of component failures on the function of superior systems are assessed. Therefore, an inductive procedure is applied when investigating the consequences of faults. The procedure is shown in Figure 1 [37].

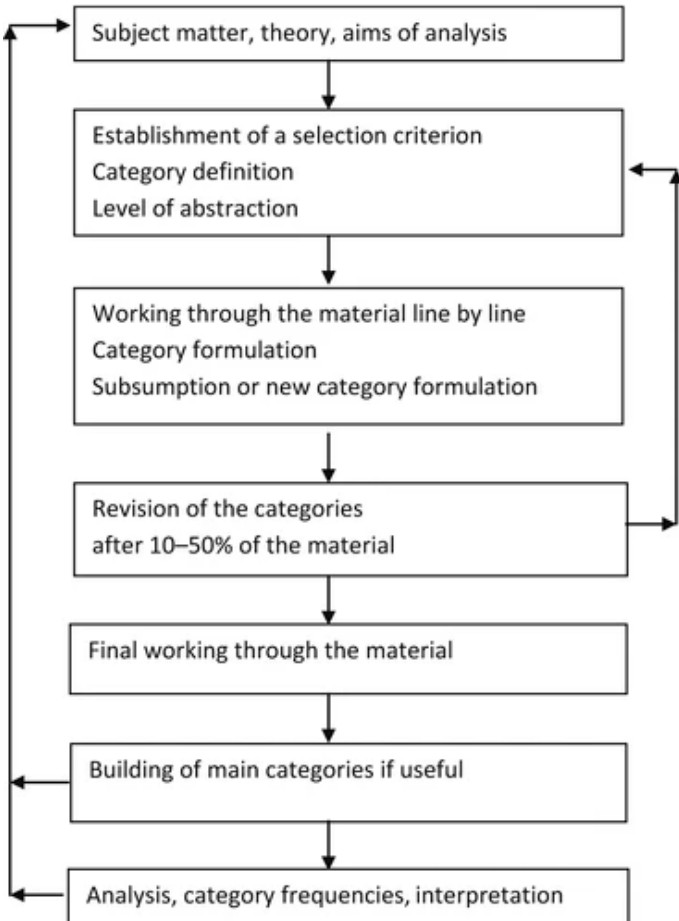

**Figure 1.** Inductive analysis procedure [38].

Among the important methods of inductive reliability analysis, we include the calculation of the prediction of failure-freeness via a calculation from the data of the element failure intensity—part of this is a method suitable for estimating the approximate intensity of system failures. It provides a prediction of system failure if there is an analysis of stress on the parts, which gives real values of the element failure intensities. The method is usually used during the early design stages. Knowledge of the reliability characteristics of system elements is a basic prerequisite for system reliability modeling. We can obtain the characteristics in several ways. The data can be obtained directly from the databases of the specific element manufacturer. In practice, the manufacturer of a specific element is unable or unwilling to provide information about product reliability. In this case, we are looking for ways to determine or estimate them. Accelerated tests are applicable, especially for electronic elements. In the pre-production phases, we estimate reliability using internationally recognized reliability databases or standard methods of calculating the numerical value of the failure intensity ($\lambda$) or the mean time between failures (MTBF).

The deductive procedure is based on the progress of the analysis from global (general) problems to elementary problems. From the analysis of system faults at the highest level of breakdown, one proceeds to the analysis of their causes and the shares of elements faults in these faults. When investigating the causes of malfunctions, a deductive procedure is then applied [39,40]. This procedure is used, for example, in fault tree analysis (FTA). A fault and the FTA procedure are shown in Figure 2.

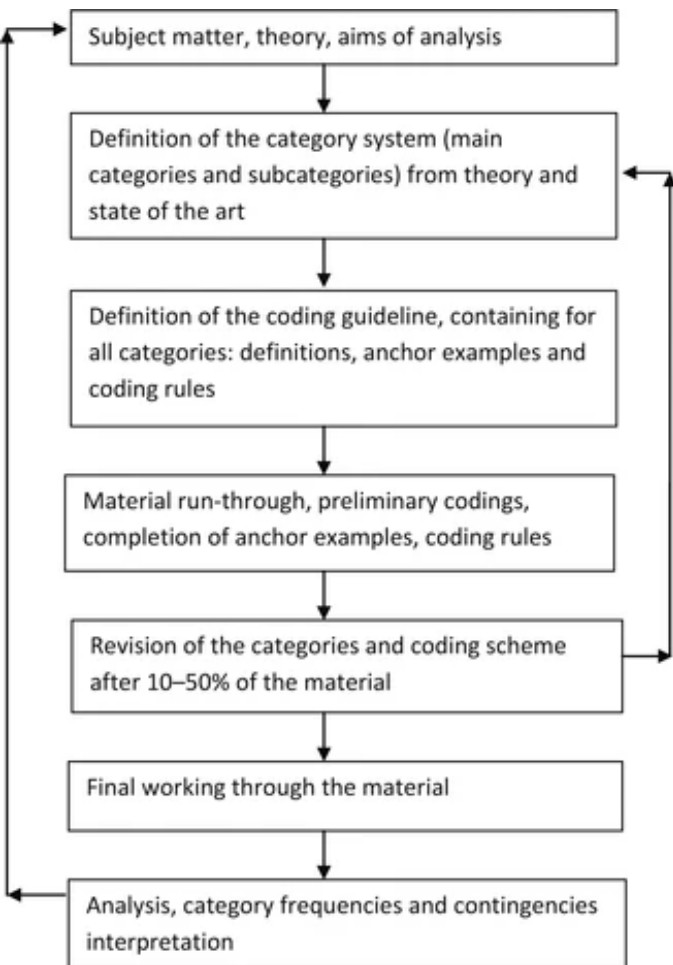

**Figure 2.** Deductive analysis procedure [38].

Other deductive methods include the RBD dependability block diagram and fault tree analysis (FTA). RBD is a basic tool for investigating reliability using a block diagram, which is a graphical representation of the system structure. It is a deductive method of determining the probability of the failure-free operation of the system from the probabilities of failure-free operation of elements. It considers the probabilities of the occurrence of phenomena in complex systems and, based on the phenomena of individual elements, expresses the reliability of the entire system. This method can only be used for systems where the failures of individual elements are independent. Probabilistic analysis via a block diagram of fault-free operation is used to predict and examine indicators of the dependability, availability, maintainability, and safety of the system. FTA is a deductive method aimed at the precise identification of causes or combinations of causes that may result in a defined adverse event. The analysis is mostly quantitative, but in the case of further use, it is also qualitative. FTA is an organized graphical expression of conditions or other factors that cause or contribute to the occurrence of a defined undesirable event, designated as a peak event—malfunction. The display of the fault tree is in such a form that it can be understood, analyzed and, if necessary, changed in order to simplify the identification of the monitored fault. Starting from the highest level of the fault tree—an undesirable condition—it is possible to proceed through the individual levels of the fault condition tree up to the elementary cause at the corresponding level of the fault condition tree. In this way, it is possible to investigate any dependencies in the system as well as its subsystems. When analyzing via this method, a systematic approach is necessary, as it is necessary to determine the functional links between the elements of the monitored system. The FTA method, traversing the tree of failure states from top to bottom, makes it

easy to recognize the causality of an undesirable state. In addition to the basic definitions, such as system, subsystem, components, it is also necessary to clarify some specifics of this method. It is necessary to consider the system and its structure, as well as its functions. One technical system can perform different functions. Individual subsystems, like the system, can have several sub-functions. We call the lowest hierarchical element of the system a component [41].

*2.8. Deterministic and Stochastic Analyses*

Deterministic model—parameters (input data) enter the model as constant values. The elements and the relationships between them are predetermined, and the behavior of the model is influenced by the specified conditions. Quantities and links between elements are deterministic and simplified. Several models of physical events that describe reality with sufficient accuracy have these properties [42].

Stochastic model—parameters (input data) enter the model based on the value of the relevant probability distribution depending on time.

Elements or relationships between them have the characteristics of the statistical evaluation of random phenomena by expressing random variables of random processes. A stochastic model considers one or more random components and approaches real events. The stochastic model does not correspond to the real situation exactly, but with a certain probability. Stochastic modelling is related to the creation and solution of stochastic models [43,44]. It has application along with possibilities to perform simulations. Using the constructed stochastic model, it is possible to simulate the course of events with different parameters of the model and observe the expected behavior of the system. Information needed for making decisions can be derived from the analysis of the model's behavior in the simulation task.

Standard IEC 300-3-1: 1993—"Reliability management—Part 3: Instructions for use. Section 1: Reliability analysis methods. This set of methodological instructions deals with the general principles of approaches when performing reliability analyses and in this context expresses the general relationships of the methods to the general procedure reliability analysis of objects. It also presents the characteristics of the most frequently used qualitative and quantitative reliability analysis methods. Based on this information and subsequent (or related) standards, it is possible to determine the possibilities of its application for individual specific objects for each individual method [45].

The principle applies that qualitative methods of reliability analysis are applied in the first phase of the initial analyses of the object, as a rule, due to the fact that there is no database available to the extent that they are required for the implementation of quantitative reliability analyses. This first phase of the application, i.e., the application of qualitative methods of reliability analysis, however, serves very well to describe and define the states of the analyzed object. This will facilitate the preparation for the implementation of the second phase—i.e., performing quantitative reliability analyses.

However, as reliability has developed as a scientific field, methods of reliability analysis have also developed. Today, the most important reliability analysis methods are already standardized, and instructions for their use are available in the form of national and international standards. We can consider the following methods as the most precise and most suitable methods for reliability analysis [46,47].

*2.9. Reliability Analysis Methods*

The main steps in predictive analysis can be divided into four steps (stages):

- Functional and technical analysis;
- Qualitative analysis of predictive methods;
- Quantitative analysis of predictive methods;
- Synthesis of analysis results.

In the first stage, the first data about the system and its purpose, target properties, and functional and technical characteristics are collected. This information is necessary

for defining the system and its properties. Above all, it is necessary to collect as detailed information as possible about its elements, from which the system is created. The first (preliminary) functional analysis is performed, which should result in more detailed identification and definition of the system main functions. It is also important for defining all significant external limitations of functional properties and operating conditions. It is the preliminary (first) stage of the qualitative analysis which helps to complete the data needed in the next stages of the analysis, and above all, it helps to identify all functions and their limitations [48,49].

The ultimate goal of qualitative analysis is to find all failures, identify their causes, describe the consequences that may arise from failures, and specify their impact on the system function. There are many formal analysis procedures. It is up to the analyst to choose the best one for the targeted purpose, considering the materials available to it and the goals of the analysis. Qualitative analysis primarily serves to build an appropriate system reliability model. The model must be based on the structural breakdown of the system and several assumptions adopted for the solution, e.g., if it describes a catastrophic or other fault condition, to which the configuration of the system or its operating phase the model applies, which faults are considered significant or even catastrophic, or which factors significantly influence the occurrence of these faults [50]. The modelling of the system reliability is closely linked with the modelling of physical phenomena and processes (degradation processes), which can result in a certain phase of the transition to a failure state. The analyst is forced to build and verify through analysis several hypotheses and assumptions about the correct or malfunctioning function related to the analyzed system. It should be emphasized that the quality of the analysis is directly dependent on the used functional model, which must cover all significant disorders and their interrelationships as accurately as possible [51].

From the very beginning of qualitative analysis, its objectives must be clearly defined. It is necessary to find out whether a study containing the concept of reliability, the determination of reliability requirements with special emphasis on the requirements of lifetime, failure-freeness, maintainability, readiness, safety, or other reliability indicators has been processed. An important part of the qualitative analysis is the determination of the scope, aimed at the depth of the analysis. To what depth of functional breakdown will the analysis be made is usually determined by the depth and range of information available in the system and its elements and the level of the system development. The structural division of the system into elements must also be in accordance with the requirement for depth of analysis. Even if the depth of the breakdown is arbitrary, it is not expedient to carry it out to a greater depth than what specific information is available about the reliability of system elements, about possible failures, their causes, and consequences. The designation of system elements should be understood from a practical point of view as that part of the system for which analysis can be carried out, for which manifestations of faults, their causes, and consequences can be specified, and for which numerical data on faults are available. It is not always necessary to perform a reliability analysis down to the lowest level of the breakdown [52].

*2.10. Quantitative Analysis of Predictive Methods*

As part of the quantitative analysis, a calculation (estimate) of the quantitative (numerical) value of suitable selected reliability indicators is made in terms of, for example, the probability of the occurrence of a failure, or the initial severity of a failure, or another indicator [53]. The numerical value of the probability is obtained through suitable and permitted manipulation of the model and consideration of the elementary phenomena that the model structurally connects in the analyzed (undesired) failure state of the system. Apart from the permitted manipulation of the model and the correct selection of input elementary phenomena and input data, it is also necessary to properly consider the following aspects [54]:

- The duration of the process to which the analysis applies (duration of the activity, phase of the process, etc.);
- Methods of influencing the correct function of backup elements and subsystems, and the time of carrying out tests;
- Principles of preventive and corrective maintenance (frequency and duration);
- The permissible range and speed of change in operating conditions, given that the model itself and all transient quantities have an inherently stochastic nature; if it follows stochastic laws and is therefore burdened with a certain "uncertainty" in its properties, the result of the analysis will also be burdened with a certain risk uncertainty in conclusions and recommendations. The level of this risk can be reduced, but it cannot be eliminated. Uncertainties are, e.g., associated with the assessment of the consequences of element failures on the severity of the system failure, with the estimation of the probability of the element failures, with the assessment of the change in transport conditions impact on the occurrence of a failure, etc. We can assess and perform to some extent a reduction in these uncertainties by analyzing the sensitivity of the system to the mentioned random influences.

Quantitative analysis can be carried out "manually" if the systems are simple and not too extensive; otherwise, it is carried out with the help of computer technology and special software developed for this purpose.

With regard to the depth of the system division, i.e., the scope of the analysis, the used analysis method depends on the resources and information available for the analysis. If necessary, expect that only limited resources will be available. If the division of the system is carried out too deeply and the chosen methods of analysis are complex and cumbersome, the analyst may be pressed for time due to the demanding scope of the work, and the deadline for the completion of the analysis may be threatened [55].

Qualitative modelling, which is an implicit part of the analysis, has quantitative aspects. Identifications and definitions of possible faults, their manifestations, consequences, and the risk of their occurrence are always stochastic in nature and contain estimation errors. Therefore, in the analysis, we can always only assume the occurrence of faults and their consequences, usually based on the basic experience gained empirically from the operation of the same or related systems. We then transfer these experiences to the expected manifestation of the new system. At the same time, it is also necessary to consider such types of failures or even their combinations, which can only be predicted, including those that have not yet occurred and for which there is no practical experience. With them, we do not have any quantitative information about the probability of their occurrence, so we must estimate them and thus introduce additional uncertainties of a stochastic nature into the analysis. These uncertainties can eventually be corrected later on the basis of the actual operation. So, the qualitative and quantitative aspects of the analysis are closely related to each other.

The conclusions from the qualitative and quantitative analysis can clarify several aspects associated with the reliability of the system and can also correct the original division of the system into elements, in terms of their selection, their reliability properties, and their influence on the used system reliability model.

A suitable method that analyzes the reliability of qualitative and quantitative methods is the FMEA method—failure mode and effect analysis. It is a method of reliability analysis that facilitates the identification of faults with significant consequences affecting the function of the system in the considered application. In general, failures or failure modes of any element negatively affect the function of the system. In the analysis of system integrity, safety, and readiness, a qualitative analysis on the one hand and a quantitative analysis on the other, which complement each other, are required. Quantitative analysis methods allow the calculation or prediction of system indicator operability when performing a certain task or during longer operation under specific conditions. Typical indicators express operation dependability, safety, promptness, failure intensity, mean time to failure, and others. The FMEA method is based on a certain design level, e.g., element or subsystem levels for which

failure criteria (primary failure modes) are available. It is based on the characteristics of the basic element and the functional structure of the system and determines the relationships between element failures and system failures, function failure, operational limitations, and the degradation of serviceability or system integrity. To evaluate the second- and higher-order faults for the system and subsystem, in some cases, it is also necessary to consider the sequence of phenomena in time.

Strictly speaking, FMEA is limited to the qualitative analysis of the material object failure modes and excludes human and software errors, even though both types of errors usually occur in common systems. These can be included in this method in a broader sense. The severity of the failure consequences is described by criticality. There are several classes or levels of criticality depending on the hazard and impairment of the system, and sometimes also the probability of their occurrence. It is best to determine this probability separately.

## 3. Reasons for Reliability Analysis during the Product Life Cycle

The historical development of defining the reliability concept went through several stages. Reliability is a commonly used term that has been adopted for years as a quality attribute in the given field. From its beginnings in 1816—when the word reliability was first coined by Samuel T. Coleridge—reliability has grown into a ubiquitous attribute with qualitative and quantitative indicators that permeates all aspects of our current technologically demanding world [56]. After achievements through various problem-oriented efforts in the field of reliability theory of technical systems in the late 1950s, development in the field continued along two paths in the 1960s. In the first degree, there was specialization in this discipline, and in the second degree, there was a trend from component-level reliability to system-level attributes (system reliability, efficiency, availability, etc.) [57]. During the Second World War, reliability was evaluated primarily for weapon systems as the probability of whether the object would be able to perform the required functions without failure, for a specified period of time, under specified operating conditions. Above all, failure-freeness was expressed in percentages. In the 1960s, it was stated as the general ability of a product to fulfil the required functions during a specified period of time, under given operating conditions, which is expressed by partial properties such as failure-freeness, service life, maintainability, repairability, readiness, storability, and diagnosability. Reliability analysis of a technical system in the engineering segment is very important, and the advantages and benefits of such an analysis can be summarized as follows:

- The investigation of the regularity of failure occurrence, the influence of various factors on the causes of failures, and the ability to describe the reliability of the technology and its lower structural parts in mathematical terms with the aim of designing, manufacturing, and using reliable technology;
- The ability to search for ways to increase the reliability of products, during their construction and production, and search for methods for maintaining reliability during use and storage;
- The creation of methods of checking product reliability and checking reliability when taking over large quantities of production;
- The ability to deal with reliability indicators, examining their relationship to economic and efficiency indicators [58].

Reliability is therefore a general property, namely the ability of the product to fulfil the required functions within the specified time, while maintaining the operating parameters given by the technical conditions. By technical conditions, we mean a summary of specific technical and operational properties, prescribed for the required functions of the product, the method of its operation, maintenance, repairs, transportation, storage, etc.

The current approach considers that to ensure usability, it was crucial that the product perform the required functions under specific operating conditions, while maintaining the specified operating indicators throughout the period of use. It goes without saying that reliability always decreases in operation. The rate of decrease in reliability is influenced

by the level of technology operation, machines and equipment, correct use, repairs, maintenance, diagnostics, etc. In the complex care system, all measures aim to ensure that the ability of technology, machines, and equipment to fulfil the task is maximally ensured. The unreliability of the technology places higher demands on the number of security personnel, the organization of repairs, equipment, and the level of means to conduct maintenance and repairs. In operation, the reliability properties are maintained not only via a suitable method of operational use, but also through so-called comprehensive care, which includes all measures and activities of preventive and follow-up care [59].

To express the course of the disorder frequency, it is appropriate to use the so-called Bathtub curve (seen in Figure 3). The Bathtub curve of failure frequency over time is often used in the field of reliability engineering, which deals with requirements for the reliability of systems from a technical, economic, ecological, and safety point of view. The Bathtub curve consists of three parts. The first part shows a decreasing frequency of failures and expresses early failures. The second part shows a constant frequency of failures, and the third part shows an increasing frequency of failures and expresses failures due to wear. The Bathtub curve that measures the slope reveals correlations between quality and reliability. Assuming that quality and reliability share a relationship along a time–stress continuum, this relationship can be used such that current quality measures can predict future reliability experiences [60]. The name of the curve, Bathtub, is derived from the shape of a bathtub: sloping walls and a flat bottom. The graphical representation of the curve is illustrated in Figure 3. The Bathtub curve hazard function (blue, upper solid line) is a combination of the decreasing hazard of early failure (red dotted line) and the increasing hazard of wear failure (yellow dotted line), plus the constant hazard of random failure (green, lower solid line).

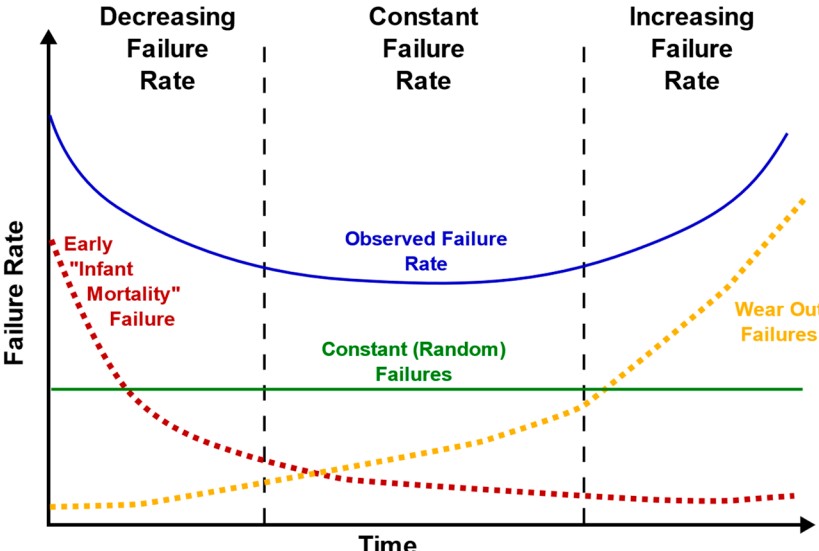

**Figure 3.** Graphical representation of the Bathtub curve.

The first part of the curve describes early failures. At this stage, a high number of failures is seen due to errors in design or manufacturing. The failure rate, however, is decreasing, because the products which have defects and are therefore failing fast are removed from the set. The middle part of the curve describes the useful lifetime of a product. During this period, the failure rate is constant. The failures seen are random failures, which can be caused, for example, due to random external stresses or mishandling of a product. The last part of the curve describes the wear-out failures of a product. At this stage, the failure rate is increasing, as the aging of components and materials is accelerating the occurrence of failures. At this stage, the failures can be caused, for example, due to corrosion, oxidation, or fatigue.

The Bathtub curve is a basic expression in reliability theory because it expresses the probability that a component will remain in a fault-free state even after a specified time.

Based on this, it is possible to estimate the availability of components and perform a reliability analysis. However, as for the reliability model of repairable components, its formulation becomes difficult because not only the curve of the tub is required, but also the speed of repair [61]. The process described in this way, which is based on a statistical expression, can be assessed theoretically using a simple predictive curve model [62,63]. As a final note regarding the Bathtub curve, it is important to notice that the curve for an actual product may look very different from the one shown in the figure. For example, there may be very few early failures, or the failures due to wear may start very early and happen slowly, causing the curve to increase slowly already during the useful lifetime. There are endless possibilities, but the simple Bathtub curve is still a good tool to describe typical failure rate behavior.

Another trend in the development of reliability analyses and definitions can be considered the current stage. It defines reliability as a collective term used to describe readiness and the factors that affect it:

- Trouble-free operation;
- Maintainability;
- Security of maintenance.

Reliability is currently referred to in English as dependability. The original term reliability is already used exclusively in standards to denote fault-freeness. Reliability assessment using degradation data has become an important approach in assessing the reliability and safety of technical systems [64].

A valid definition of reliability has the following characteristics:

- It expresses the fact that the ability of the object to fulfil the required functions is significantly influenced by internal factors, e.g., degree of assurance of the required maintenance;
- The term reliability is used only for a general description, and it is not possible to quantify and summarize it with any numerical indicator;
- It is possible to quantitatively evaluate individual factors using specific reliability indicators: promptness, trouble-free operation, maintainability, etc.

The attributes of reliability are not defined in the valid terminological standards. Nevertheless, we distinguish the following types of reliability:

- Estimated reliability: it is the reliability that is the result of calculation, analyses, and forecasts of the designed object reliability;
- Inherent reliability: it is the reliability "embedded" in the object during its design and manufacture. It does not contain the worsening effects of operating conditions, environmental conditions, maintenance methods, or the human factor;
- Operational reliability: it is the reliability considering the influence of operational and other conditions.

When examining the definitions of reliability in more detail, we find that they show two different tendencies, which express a narrower and a broader understanding of reliability:

1. Quantitative definition of reliability: when reliability is defined as a certain numerical characteristic;
2. Qualitative definition of reliability: when, more generally, in a broader understanding, product reliability is considered a property of the product that depends on fault-freeness, maintainability (maintainability and repairability), and security of maintenance of the product and its elements, which guarantees the fulfilment of the requirements for the proper functioning of the product.

However, reliability is a general property of an object that cannot be summed up and expressed by a single numerical characteristic. It can be expressed and quantified by indicators of monitored quantities of partial reliability properties. The evaluation of technology reliability consists of selecting the numerical values of the reliability indicators and comparing them with the determined values or the values of the comparable technology indicators. One of the key steps has been the development of reliability indicator modelling

techniques that enable the fast, cheap, and accurate evaluation of engineering systems. The modelling of engineering systems has gradually become more accurate, but also more complex, with increasing computational availability [65].

However, the qualitative expression of individual properties is not enough, and their quantification is important, so reliability can be expressed as any physical quantity. For this reason, we use reliability indicators in practice, which are quantitative characteristics of one or several properties that comprise reliability and which have an exact mathematical expression for specific quantities monitored through deterministic or stochastic relationships. In order to be able to express the reliability value of a technical system and to apply the steps of reliability analysis, we need to know what monitored variables and parameters comprise the resulting reliability. In the following chapter, the research is processed on monitored variables and reliability indicators that enter a complex analysis of the technical system reliability.

## 4. Observed Reliability Variables

Quantities monitored in reliability are closely related to the random occurrence of monitored phenomena. Thus, quantities associated with time, the progress of operating units, the number of failures, etc., are monitored. Indicators in the description of reliability, on the other hand, in the probabilistic sense, mean a function or value used to describe a random variable or a random process. Interrelationships between properties, monitored quantities, and reliability indicators are defined in the following chapter.

### 4.1. Dependability

Dependability is the ability of an object to perform the required functions under specified conditions in a given time interval [66]. The specified conditions are expressed by prescribed modes of operation, assumed operating conditions, and a specified system of complex care. It is assumed that the object at the beginning of the interval is able to fulfil the required functions [67]. This is also how reliability is sometimes defined in the older literature—we usually come across the term reliability, while it is now called dependability [68]. Machines and equipment contain several elements and higher units that are not repaired in the event of a malfunction but are replaced with new ones. Therefore, we must distinguish which object we are examining from the point of view of dependability. These are objects and their properties characterizing the ability to carry out repairs [69]. A repairable object is an object that can be repaired after a failure. A repairable object can be restored or not restored according to the nature of the operation. An unrepairable object is an object that cannot be repaired after failure. An unrepairable object is therefore always unrestored. In the case of unrenovated objects, the observed variable is characterized by the time (time interval) of the object's operation until the occurrence of the first failure (abbreviated time to failure). Time means the time interval required for the object to perform a predetermined activity. It is given numerically in units of time, or another quantity that can be converted to time (number of kilometers travelled, cycles, switches on, lifts, revolutions, amount of work—so-called operational units). The object can work intermittently or continuously—only the net operating time is considered. In the case of renovated objects, all operating times are monitored, i.e., the time until the first failure appears, and also the time between subsequent failures.

Dependability indicators are functions or numerical values of a random variable. We distinguish dependability indicators for restored and non-restored objects. The indicated indicators can be found in Tables 1 and 2.

**Table 1.** Selected dependability indicators of non-restored objects.

| Monitored Value | Dependability Indicators | Marking |
|---|---|---|
| Time of operation until failure | Probability of operation dependability | $R(t)$ |
| | Intensity of disturbances | $\lambda(t)$ |
| | Mean time to failure | $MTTF$, $t_1$ |
| | $p$-quantile of time to failure | $T_{1p}$ |

**Table 2.** Selected indicators of dependability of restored objects.

| Monitored Value | Failure-Free Indicators | Marking |
|---|---|---|
| | Probability of operation dependability | $R(t_1, t_2)$ |
| | Intensity of disturbances | $\lambda(t)$ |
| | Medium intensity of disturbances | $\lambda(t_1, t_2)$ |
| Time of operation until failure | Fault current parameter | $z(t)$ |
| | The mean parameter of the disturbances | $z(t_1, t_2)$ |
| | Mean time of operation until failure | $MTBF, t$ |
| | $p$-quantile of time between disturbances | $t_p$ |

Probability of survival: the probability that the object will not fail in a given time interval. It expresses the probability that the object can perform the required function under the given conditions in the given time interval $(t, t_1, t_2)$ [70].

$$R(t) = 1 - F(t) = exp(-\lambda t), \lambda > 0, t \geq 0, \tag{1}$$

Failure rate: expresses the limit, if there is one, of the ratio of the conditional probability that the time instant of the occurrence of the failure of the object lies in the given time interval $(t, t + delta\ t)$ to the length of the time interval delta $t$, if delta $t$ is close to zero, provided that the object is in a usable state at the beginning of the time interval [71].

$$\lambda(t) = \frac{f(t)}{R(t)} = \frac{\lambda exp(-\lambda t)}{exp(-\lambda t)} = \lambda = \text{konst.}, \lambda > 0, t \geq 0, \tag{2}$$

Mean failure rate: the mean value of the instantaneous failure rate in a given time interval $(t_1, t_2)$ [72].

Mean time to failure: this expresses the expected time until failure.

$p$—time to failure quantile—guaranteed time of trouble-free operation $t_p$.

$$\bar{t} = MTTF = \frac{1}{N}\sum_{i=1}^{N} t_i = \frac{\sum_{i=1}^{N} TTF}{N}, \tag{3}$$

where the following variables are defined:

$t_i$—the time until failure of the *i*-th monitored product;

*TTFi*—the time until failure of the *i*-th object.

Failure intensity: the limit, if any, of the ratio of the average number of failures of the repaired object in the time interval $t + delta\ t$ to the length of this interval delta $t$, if the length of the time interval approaches zero.

$$Z(t) = \lim_{\Delta \to 0} \frac{E[Z(t + \Delta t) - Z(t)]}{\Delta t} = \frac{dZ(t)}{dt}, \tag{4}$$

where the following are defined:

$E[\ ]$—the mean value symbol;

$Z(t)$—the expected number of failures in the time interval $(0, t)$.

Mean failure intensity: mean value of the instantaneous parameter of the fault current in the given time interval $(t_1, t_2)$, [73,74].

*4.2. Maintainability*

By maintainability indicators, we generally mean a function or a numerical value used to describe the probability distribution of a specific monitored (random) variable that characterizes the maintainability of an object. Maintainability refers to the ease and speed with which maintenance work can be carried out. It relates to the processes of securing personnel, technical means, materials, and maintenance technology, and their optimization. When a product has a reasonably long technical life, the costs of operation and security of

its operation during its lifetime can significantly exceed the initial investment costs [75]. Efforts are made and expenses incurred to achieve the readiness of the product as one which is as easy and cheap to maintain as possible, leading to very significant life cycle cost savings. Product maintenance costs depend on how many products are used. For many devices used for a long time, even a small improvement in maintainability can result in significant cost savings. When a product is sold on the open market, the concept of easy maintenance at low cost is of great importance in the customer's decision to purchase equipment with low operating costs, and thus, providing maintenance and ensuring the reliability of objects (products, equipment, and systems) throughout their entire life cycle is a key element. The required functionality, capability, and reliability is achieved by providing the necessary maintenance and assurance of maintenance, appropriate design, quality of manufacture, and good operating instructions and procedures. In terms of the scope and type of maintenance, the provision of maintenance depends on the needs of the customer, the nature and condition of the object, the required readiness, and other factors. If these factors change, especially during the operation and maintenance phase, maintenance and maintenance assurance may need to be adjusted. Inadequate, excessive, or incorrect maintenance can cause breakdowns that can significantly reduce the availability of facilities and lead to significantly increased costs due to the loss of performance and possible subsequent damage. Reduced availability due to poor maintainability and maintenance assurance leads to downtime losses resulting in lost revenue that can be greater than the maintenance costs or even acquisition costs. It can affect safety, and in some industries, it can be the most significant driver of losses [76].

Therefore, maintainability is the object's ability to determine the real state, causes of malfunctions, prevention, elimination of their consequences through prescribed diagnostics, and preventive maintenance and repair. It is a feature expressing the ability to carry out routine maintenance, repairs, and diagnostics expressed by the terms according to the older literature—maintainability, repairability, and diagnosability. From a qualitative point of view, it expresses operational reliability expressed by the efficiency, economy, ease of maintenance, level of complex design features, and characteristics of the equipment, which allows maintenance to be carried out by qualified workers under the given conditions of environmental influences, operating conditions, and the human factor [77].

The relationships between reliability and maintainability can be expressed quantitatively. Maintainability is increased by using effective diagnostic methods and through the organization of maintenance, thus ensuring maintenance [78]. Reliability is increased by the possibility of preventive interventions, or by the timely replacement or repair of parts, undemanding maintenance, and quick and high-quality maintenance [79].

The main factors affecting the maintainability of the product are mainly these:

- The structural complexity of the product and its groups;
- Spatial arrangement of the product and access to groups during diagnostics, maintenance, and repairs;
- Complexity, interchangeability, and replaceability of parts, the unification of parts, and the technical equipment for performing diagnostics, maintenance, and repairs;
- Diagnostic, maintenance, and repair procedures;
- Qualification of managers, operators, and workshop specialists;
- Technical equipment and flexibility not only of the user but also of the organizations that provide the user with supplier maintenance.

From the above definitions, it is clear that the maintainability of the machine or equipment is determined by the design of the structure, which conditions low demands on the volume of preventive maintenance and facilitates the execution of repairs and a whole set of operational (logistical) measures which include qualified maintenance personnel, an optimal system of maintenance and technical documentation, test and maintenance equipment, and the supply of spare parts, maintenance objects, and their equipment [80].

All participants in the life cycle participate in ensuring maintainability, repairability and diagnosability—suppliers and customers in the following positions:

- Manufacturers;
- Suppliers;
- Users.

The product must be designed, manufactured, and operated so that it can be easily maintained while ensuring minimum costs, i.e., maintained, repaired, diagnosed, and scrapped.

Examples of practical signs, measures, and situations associated with maintainability are provided below:

- Maintainability requirements are stated in specifications and standards;
- Suitably oversized and correctly structurally designed and manufactured products [81].

The methodical answer to most of these problems is provided by the gradually introduced and relatively extensive, relatively complex standard STN IEC 706 Guidelines for the maintainability of equipment, which consists of nine sections. A summary of the sections according to the standard is presented in Table 3. Indicators of maintainability and assurance of maintenance of non-renewable objects are found in Table 4. Indicators of maintainability and assurance of maintenance of restored objects are found in Table 5.

**Table 3.** Sections of STN IEC 706.

| | |
|---|---|
| 1. Introduction to maintainability<br>2. Maintainability requirements in specifications and contracts<br>3. Maintainability program | STN IEC 706-1 |
| 4. Diagnostic testing | STN IEC 706-5 |
| 5. Sustainability studies at the design stage | STN IEC 706-2 |
| 6. Verification of maintainability<br>7. Collection, analysis, and presentation of data related to maintainability | STN IEC706-3 |
| 8. Maintenance planning and its provision | STN IEC 706-4 |
| 9. Statistical procedures in maintainability | STN IEC 706-6 |

**Table 4.** Selected indicators of maintainability and for ensuring the maintenance of non-renewable objects.

| Monitored Values | Reliability Indicators | Marking |
|---|---|---|
| Time of preventive maintenance | Average preventive maintenance time<br>Probability of performing maintenance | $\bar{t}_{0m}$<br>$M(t_1, t_2)$ |

**Table 5.** Selected indicators of maintainability and for ensuring the maintenance of restored objects.

| Monitored Values | Reliability Indicators | Marking |
|---|---|---|
| Maintenance time<br>Time of preventive maintenance | Probability of performing maintenance<br>Average preventive maintenance time | $M(t_1, t_2)$<br>$\bar{t}_{0m}$ |
| Repair time | Repair intensity<br>Medium repair intensity<br>Average repair time<br>$p$-quantile of repair time | $\mu(t)$<br>$\mu(t_1, t_2)$<br>$MRT, \bar{t}_{0m}$<br>$\bar{t}_{po}$ |
| Labor of maintenance | Medium labor of maintenance | $\bar{t}_{pm}$ |
| Maintenance time after failure | Average time to recovery | MTTR |

By maintainability indicators, we generally mean a numerical value or function used to describe the probability distribution of a specific monitored (random) quantity that characterizes the maintainability of an object.

Average maintenance time $\overline{t_U}$

It is given by the arithmetic mean of the measured maintenance times of the object for the observed period and is expressed as:

$$\overline{t_U} = \frac{\sum t_{iU}}{N},$$ (5)

where the following are defined:

$t_{iU}$—the maintenance period of the *i*-th object;
$N$—the total number of objects in operation in the monitored period.

Probability of performing maintenance $F(t)_U$

It is given by the probability of completing the maintenance of the object by a given time after the start of the maintenance and is expressed by the relation:

$$F(t)_U = \frac{\Delta n_U}{n_U},$$ (6)

where the following are defined:

$\Delta n_U$—the number of objects renewed (maintained) in the operation interval $0 \to t$,
$n_U$—the total number of objects requiring maintenance at the beginning of the monitored interval.

Maintenance intensity $\lambda(t)_U$

It is given by the probability of maintenance completion for an infinitesimally small time unit after a given moment with the condition that maintenance has not been completed by this moment. It is determined by the ratio of the object number restored for a small time unit after a given moment to the object numbers that are being renewed (maintained) at this moment. A relationship applies:

$$\lambda(t)_U = \frac{\Delta n_U(\Delta t \to 0)}{n_U(t)},$$ (7)

where the following are defined:

$\Delta n_U(\Delta t \to 0)$—the number of restored objects in the interval $(\Delta t \to 0)$ after time *t*;
$\Delta n_U(t)$—the number of restored (maintained) objects at the beginning of the interval $\Delta t$

Maintenance factor $K_U$

It is given by the ratio of the number of hours spent on maintenance and routine repairs to the time of its trouble-free operation and applies in the equation:

$$K_U = \frac{t_U}{t_p},$$ (8)

where the following are defined:

$t_U$—the number of hours for maintenance and routine repairs during the monitored period;
$t_p$—the number of hours of trouble-free operation (operation) for the monitored period.

The maintenance factor is advantageously used for planning the time (capacity) required for maintenance and repairs.

Frequency of maintenance $K_W$

It is the ratio of maintenance and normal repairs of the object to the time of its forced decommissioning and the time of fault-free operation during a certain period, and the following applies:

$$K_W = \frac{n + m}{t_p + t_U},$$ (9)

where the following are defined:

$n$—the number of individual maintenances of the object during the monitored period;
$m$—the number of common repairs for the monitored period.

The greater the value of the maintenance frequency at the same value of the maintenance factor, the shorter the continuous operation time until the failure occurs, or until the preventive maintenance of the vehicle is carried out.

### 4.3. Additional Definitions and Terminological Concepts from the Field of Maintainability

Maintenance time: the time interval during which the maintenance intervention is carried out either manually or automatically, including technical and logistical delays. In the event that logistical delays are not considered, but only the part of the maintenance period when the maintenance intervention is carried out, we speak of the period of active maintenance.

Preventive maintenance time: the part of the maintenance period during which preventive maintenance is carried out on the object, including technical and logistical delays contained in preventive maintenance. If logistical delays are not considered, we are talking about the period of active preventive maintenance, which we denote using the symbol TPU [82].

Corrective maintenance time: the part of the maintenance period during which post-failure maintenance is performed on the facility, including technical and logistical delays inherent in post-failure maintenance. If logistical delays are not considered, we are talking about the period of active maintenance after a breakdown.

Repair time: the part of the active maintenance period after a breakdown during which repair work is carried out on the object. The repair time can be further divided into the time of localization of the damaged part, the time of the active repair during which the own repair operations are carried out, and the inspection time when the function of the object is checked. We indicate the repair time with the symbol too.

Mean time to repair MTTR: the average time from the beginning of the repair to the moment of restoration or operability of the object.

Repair rate: the limit, if there is one, of the ratio of the conditional probability that the maintenance intervention after the failure ends in the time interval ($t$, $t$ + delta $t$) to the length of the time interval delta $t$, if delta $t$ approaches zero, provided that this operation has not ended before the beginning of the time interval.

Mean repair rate: the mean value of the immediate intensity of the correction in the given time interval.

Maintenance man-hours: the cumulative duration of individual maintenance periods, expressed in standard hours, used by all maintenance workers for a given type of maintenance intervention, or during a given time interval.

### 4.4. Ensuring Maintenance

In modern manufacturing systems, equipment experiences a continuous decline as a result of numerous internal and external factors. If timely maintenance of the equipment is not carried out prior to its failure, it will inevitably result in substantial financial losses and pose safety hazards. Preventive maintenance is a widely used and more flexible maintenance method than corrective maintenance, especially in the face of increasingly complex, sophisticated, and integrated production equipment [1].

Maintenance assurance indicators generally mean a function or numerical value used to describe the probability distribution of a specifically monitored (random) variable that characterizes the object's maintenance assurance [83]. Such a random quantity is usually the time of logistical or administrative downtime (delays).

It is characteristic of maintenance assurance indicators that they usually do not describe the properties of the object as such (its inherent properties), but most of them are primarily influenced by the applied maintenance concept, supply system, administrative procedures, etc. [84]. Selected indicators for the maintenance area include the following:

- Up time: the time interval during which the object is in a usable state.
- Down time: the time interval during which the object is in an unusable state.
- Logistical delay: the cumulative time during which maintenance operations cannot be performed due to the necessary acquisition of maintenance resources, excluding

administrative delay. Logistical delays can be caused by, for example, waiting for spare parts, experts, test equipment, and information, and unsuitable environmental conditions, etc. We refer to the logistical delay as tL.

- Technical delay: the cumulative time required to perform auxiliary technical operations relating to the maintenance intervention. We associate technical delays exclusively with maintenance after a breakdown. It expresses, for example, the time required to move the object to the relevant repair workplace and back, to clean the object before starting the repair, etc.
- Mean administrative delay: the expected administrative delay. Designation: MAD;
- Mean logistical delay: the expected logistical delay. Designation: MLD;
- *p*-fractile logistical delay: the *p*-quantile value of the logistical lag.

### 4.5. Availability

The reliability of repaired objects is primarily characterized by indicators of readiness, which comprehensively describe the fault-freeness and maintainability of repaired objects. By readiness indicators, we generally understand this to mean a function, or a numerical value, used to describe the probability distribution of a specifically monitored (random) quantity that characterizes the readiness of an object. Such a random variable can be, for example, the state of an object that changes randomly over time. For renewable systems consisting of several subsystems or elements, as a rule, the resulting level of reliability must be determined by the characterized system readiness [85]. Availability is a criterion for restored objects that assesses dependability, maintainability, and comprehensive maintenance [86]. Availability is the ability of an object to be in a state capable of performing the required function under given conditions, at a given time moment or interval, provided that the required external resources are provided [87]. External means are the means of comprehensive care.

Availability is expressed by the following quantitative indicators in Table 6.

**Table 6.** Selected availability indicators of renovated objects.

| Monitored Values | Reliability Indicators | Marking |
|---|---|---|
| Time of usable condition | Average useful life | MUT |
| | Average time duration of unusable state | MDT |
| | Immediate emergency | A(t) |
| | Coefficient of asymptotic availability | A |
| Time duration of unusable state | Coefficient of medium alertness | $A(t_1, t_2)$ |
| | Immediate emergency | U(t) |
| | Coefficient of medium unpreparedness | $U(t_1, t_2)$ |
| | Coefficient of asymptotic failure | U |

The starting point for the design of the system availability model is the so-called analysis of states in which the system can occur. The system can be in many and different states, each of which is determined by a certain combination of individual element states. Likewise, every system element can also occur in different states that randomly alternate. The process when the states of monitored objects change randomly over time is called a Markov random process [88].

Most often, states in mechanical systems are expressed via a two-state model. Depending on the state of the individual elements, the system can be either functional or non-functional. If the transitions between these states randomly alternate and can occur at any moment in time, this random process is called a simple recovery process (Figure 4).

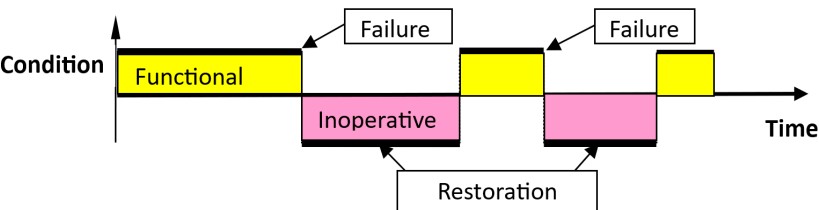

**Figure 4.** Simple recovery process.

Types and Indicators of Availability

The availability indicator is a function or numerical value used to describe the probability distribution of a specific monitored (random) variable that characterizes the availability of an object. As a rule, such a quantity is the state of the object, which randomly changes over time. Indicators of readiness are different; they differ by what kinds of states of the object we consider during the analysis. The distribution of standby times is shown in Figure 5 [89].

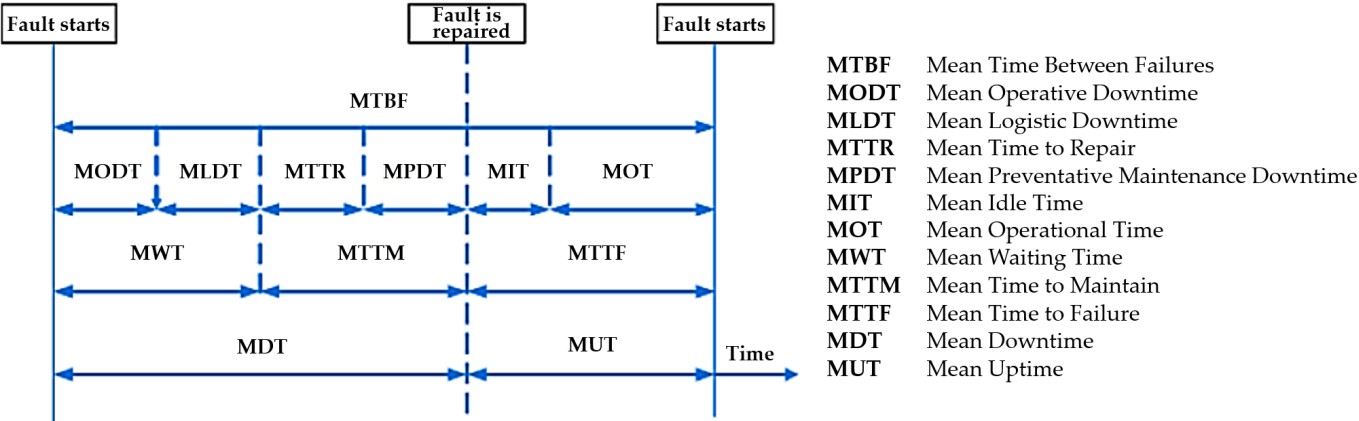

| | |
|---|---|
| **MTBF** | Mean Time Between Failures |
| **MODT** | Mean Operative Downtime |
| **MLDT** | Mean Logistic Downtime |
| **MTTR** | Mean Time to Repair |
| **MPDT** | Mean Preventative Maintenance Downtime |
| **MIT** | Mean Idle Time |
| **MOT** | Mean Operational Time |
| **MWT** | Mean Waiting Time |
| **MTTM** | Mean Time to Maintain |
| **MTTF** | Mean Time to Failure |
| **MDT** | Mean Downtime |
| **MUT** | Mean Uptime |

**Figure 5.** Terminology of times in alertness assessment.

Instantaneous or Point Availability, $A(t)$: The probability that the object (element, system) occurs in a given state at a given moment in time is described by the function of immediate readiness $A(t)$ for an operational state or by the function of immediate non-availability $U(t)$ for a non-functional state (incapable of operation). Functions $A(t)$ and $U(t)$ are mutually complementary, and the sum of their values at a given moment is equal to 1 (the probability that the object will occur in one or the other state is equal to certainty). The immediate availability function $A(t)$ expresses the probability that the object is in a state capable of performing the required function under the given conditions and at the given moment of time, provided that the required external resources are provided [90].

The conceptual model of the calculation of immediate readiness is as follows:

1. The object works correctly—it is fault-free in the period from 0 to $t$ with the probability of fault-free operation $R(t)$;
2. Or it works correctly, because after the last repair at the time $u$ $0 < u < t$ with probability $m(u)$, it is restored, which is expressed in the relation:

$$\int_0^t R(t-u)m(u)du, \tag{10}$$

where the following is defined:

$m(u)$—the system renewal density function;

3. The instantaneous standby function $A(t)$ is a summation of these two probabilities from points 1 and 2.

$$A(t) = R(t) + \int_0^t R(t-u)m(u)du, \tag{11}$$

Average Uptime Availability (or Mean Availability), $\overline{A}(t)$: Also called the average, it is expressed by the coefficient of the average readiness, which expresses the average value of immediate readiness in a given time interval $(t_1, t_2)$:

$$\overline{A}(t_1, t_2) = \frac{1}{t_2 - t_1} \cdot \int_{t_1}^{t_2} A(t) \cdot dt, \tag{12}$$

Asymptotic (steady) availability: The coefficient of asymptotic (steady) standby represents the limit of the instantaneous standby function for $t \to \infty$.

$$A = \lim_{t \to \infty} A(t), \tag{13}$$

The coefficient of asymptotic availability $A$ can be expressed by the expression:

$$A = \frac{MTBF}{MTTR + MTBF}, \tag{14}$$

where the following are defined:

*MTBF*—the mean time between failures;
*MTTR*—the mean repair time.

A simple description of the recovery process for calculating the asymptotic availability factor is shown in Figure 6.

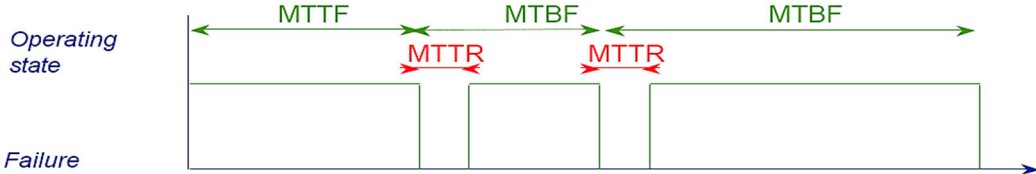

**Figure 6.** A simple recovery process for calculating the asymptotic availability factor.

It expresses the probability that the object, which is in the stable operating mode "operation—restoration", will be in an operable state at any moment in time (outside the planned period during which the use of the object is not considered, e.g., planned preventive maintenance) [91].

In technical practice, the asymptotic availability coefficient A is used very often for a stabilized recovery process under the following assumptions (the onset of asymptotic alertness is shown in Figure 7):

- The logistical, administrative, and technical delays are negligible;
- The distributions of the random variable for dependability with the parameter $\lambda$ and maintainability with the parameter $\mu$ are exponential.

If the distribution of the times between failures and the time until restoration has an exponential character, we can express the coefficient of asymptotic availability of the object:

$$A = \frac{\mu}{\lambda + \mu}, \tag{15}$$

where the following are defined:

$\lambda$—the intensity of failure;
$\mu$—the intensity of maintenance.

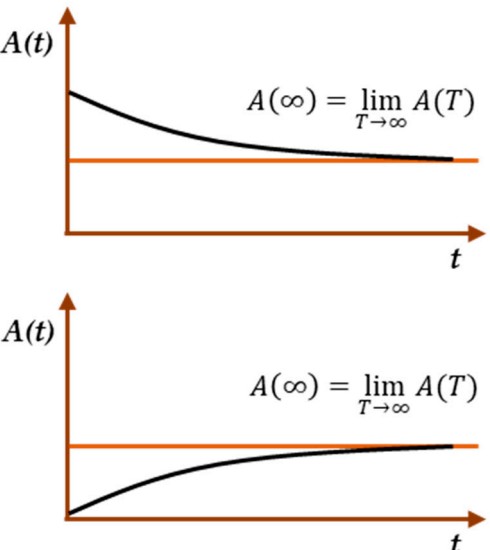

**Figure 7.** Asymptotic steady-state ramp-up at initial state 1 and 0.

Operational Availability: the probability that the device or system will satisfactorily perform the required functions under the specified conditions at any moment during the required operating time. This time includes the operation time, maintenance time after failure and preventive maintenance, administrative delay time, and logistical delay time [92].

The coefficient of operational readiness expresses the ratio of the total time of operation in a usable state to the total time including downtime.

$$A_O = \frac{MUT}{MUT + MDT},\tag{16}$$

where the following are defined:

*MUT*—mean useful time;
*MDT*—mean time duration of unusable state.

Achieved readiness: The achieved readiness factor is expressed using the mean time between MTBM maintenance and the mean maintenance downtime $\overline{M}$:

$$A_A = \frac{MTBM}{MTBM + \overline{M}},\tag{17}$$

It expresses the probability that the object will work satisfactorily at a given moment if personnel, tools, spare parts, etc., are immediately available. It excludes logistical delays, administrative and other downtimes. It contains only active preventive and corrective maintenance periods. The availability achieved is primarily due to the construction of the device and its maintainability. Equipment maintenance assurance assumes that spare parts and personnel are 100% available without delay.

For the overall description of the object availability, the most suitable of the listed indicators is the coefficient of asymptotic availability, which characterizes a certain steady level of readiness, to which the object gradually approaches with an increasing operating time.

All other statistical models created on the basis of stochastic principles always lead to a non-constant contingency function $A_i$, i.e., to the standby function depending on the time of operation $t$.

The instantaneous availability function expresses the probability that the object is in a state capable of performing the required function under the given conditions and at the given moment of time, provided that the required external resources are provided. Designation: $A(t)$.

Instantaneous unavailability function: expresses the probability that the object is not in a state capable of fulfilling the required function under the given conditions and at the given moment of time, provided that the required external resources are provided. Designation: $U(t)$.

Coefficient of mean availability: the mean value of immediate availability in a given time interval $(t_1, t_2)$.

Coefficient of mean unavailability: the mean value of immediate unavailability in a given time interval $(t_1, t_2)$.

Asymptotic availability factor (asymptotic availability): the limit of immediate availability for modelling purposes, if it exists, if time approaches infinity.

Asymptotic unavailability factor: the limit of instantaneous unavailability for modelling purposes, if it exists, if time approaches infinity.

Mean up time: the expected up time. Designation *MUT*.

Mean down time: the expected time duration of unusable state. Designation: *MDT* [93].

*4.6. Durability*

Durability is the ability of an object to fulfill the required functions, with a specified system of complex care (mainly use, maintenance, and repairs) until reaching a limit state, which can be characterized by the end of its useful life, and its unsuitability for reasons of economic, technical, or other serious factors [94]. The lifetime of objects is limited by a limit state, which we understand as a state at which further use must be interrupted for non-removable objects under the following conditions:

- Exceeding the prescribed values of technical parameters;
- The violation of safety regulations;
- A reduction in operational efficiency below an admissible acceptable value.

Durability is the cumulative value (sum) of all periods of partially operable states of the object until the moment when it will no longer be able to perform its function because it has reached the limit state. In the case of non-renewable objects, we consider the occurrence of a malfunction or the achievement of the parameter limit value, which characterizes the operability of the product, as a limit state [95].

We recognize different types of durability of objects:

Physical durability: The limit state can be the occurrence of a malfunction or the achievement of the selected parameter limit value. In the event of a malfunction, these can be as follows:

- Irreparable failures ending the technical life of the equipment, for example, destruction of the technology, which usually occurs as a result of an accident or extraordinary event;
- Repairable failures that require repairs of a higher type and which end the technical life of the technology [96].

In cases of parametric determination of the limit state, these are the limit values of the parameters that characterize the degradation of the technique either directly or indirectly and can be as follows:

- Functional parameters (power, acceleration, revolutions, flow rates, pressures, absorbed energy, efficiency, etc.);
- Direct parameters of degradation (e.g., rate of mechanical wear of parts, change in the composition of the working medium, size of the fatigue crack, etc.);
- Fault current parameter;
- Operating time (number of working hours, cycles, etc.) or age of the product.

Economic durability of technology: It is expressed by the period of economically optimal life. It can be determined by the state when the relative costs spent on repairs of equipment begin to exceed the relative costs of securing new equipment and its repairs. This type of durability can also be attributed to cases where technical repairs cannot be carried out due to a lack of spare parts (it is not practical to produce them).

Moral durability of technology: It ends when the utility value of the product (e.g., efficiency, performance parameters, etc.) has decreased compared to comparable products, or products corresponding to current scientific and technical knowledge, so that it is necessary to replace it with new technology. It is necessary to realize that the overall durability of machines and equipment is affected by the condition of individual machine parts and functional surfaces. Changes in the technical state of the art are always reflected on elementary surfaces, which are the basic object of durability research [97]. When taking a closer look at the causes of wear and technology breakdowns, it is clear that they manifest themselves in two basic forms:

- Gradually increasing degradation (corrosion, wear);
- Sudden degradation (material failure, overload).

Both types of degradation influence the occurrence of faults and their impact on the technical life of the components. In the operation of technology, there is more often a gradual approach to the limit state. Sometimes, for determining durability, reaching the limit state is not even allowed, because the resulting damage could have dangerous or economically unbearable consequences. Selected life indicators are shown in Table 7 [98].

**Table 7.** Selected durability indicators.

| Monitored Quantity | Durability Indicator | Designation |
|---|---|---|
| Period of useful (technical) life | Average useful (technical) life | $\bar{t}_l$ |
| | $p$-quantile of useful (technical) life | $\bar{t}_{tl}$ |

However, technical durability is limited by the durability of one or several limiting elements. These elements, which determine the durability of the final product, are usually the main functional and most expensive part of the product, in which the criterion for determining the durability is usually the safety of operation [99]. Determining the durability parameters of repaired products is a difficult task, which is basically solved in two ways:

The monitored value of durability is the period of useful (technical) life: Lifetime indicators express either the period of use (years, months, days) or the performed activity expressed by the consumption of operating units (running km, Mh, cycles).

They often refer only to different limit states:

Average useful (technical) life $\bar{t}_{\check{z}}$: This characterizes the period of operation (consumed operating units) of the technology until the moment of the limit state occurrence defined in the technical conditions. The length of the technical life specified in time or other units is usually stated in the technical documentation of the technology.

In statistical terms, technical life until overhaul is represented as:

$$t_{\check{Z}GO} = \frac{1}{N_0}\sum_1^{N_0} t_{iGO},\tag{18}$$

where the following are defined:

$N_0$—the total number of monitored pieces of equipment;
$t_{iGO}$—the time of the $i$-th technique after reaching the limit state (*GO*).

$p$-quantile of useful life (gamma percentage technical life) [$t_{tl}$, ($p = 100 - \gamma$)]: This is the total time of operation of the technology until its retirement or repair, which the technology will probably exceed [100,101].

We carry out the examination of the durability of the technology and its components in practical conditions with the following aims:

- Verify the quality of the production or the repair process;
- Ascertain the basis for optimization calculations and determine standards;
- Determine operational reliability and operating time;
- Detect treatment cycles, diagnostic signals, and parameters;
- Determine the consumption of spare part stocks;

- Determine operation standards with regard to repair capacity and replenishment of technology from new production [102].

*4.7. Repairability*

Repairability is a feature of an object comprising the ability and possibility to detect malfunctions and eliminate their consequences by carrying out repairs. Analogously to maintainability, repairability expresses the time, material, work, and economic demands, or conversely, how easily, undemandingly, and simply the product can be returned to a fault-free state via repair. By the term repairability, we understand it to mean the adaptation of machines and their elements to the execution of planned technical maintenance, the elimination of operational faults, and the execution of repairs [103]. Repairability is a technical–economic characteristic of the reliability of a machine or group. The costs incurred to operate the machine depend on the level of repairability.

The level of machine repairability can be increased via the following actions:

- Maximum unification and standardization of machines and their elements (fast-wearing parts and functional pairs, connecting parts, rolling and sliding bearings, detachable joints, hoses, pipelines, etc.);
- Fast and reliable evaluation of the machine's technical condition and forecasting of its remaining technical life using methods without disassembly diagnostics (a set of reliable control and signaling devices);
- Increase the adaptation of the machine to repair;
- Increase the technical life before the first repair and between repairs throughout its technical life;
- Reduce time consumption, and work and repair costs (facilitate and simplify the technological process of disassembly and assembly, make centering holes for the basic parts that are intended for renovation, increase the strength of the parts that are renovated to the repair size, increase the number of parts that will have repair dimensions, increase the possibility of rebuilding or turning parts with one-sided wear, achieve complete interchangeability of the same parts, etc.);
- Improve the design of the machine by increasing the independence of individual subgroups and groups and the possibility of replacing them with minimal work and time consumption (make covers and cabinets that ensure the disassembly of a subgroup or group so that they are structurally an independent unit, make basic surfaces or groups without additional centering and adjustment, and others) [104].

In the case of repairability, the indicators describe the random repair time, i.e., the maintenance period after failure without logistical, administrative, and technical delay, which includes location, active repair, and inspection. In general, however, we must incorporate into the recovery (repair) process the following steps:

- Identification, determination, and localization of the fault;
- Determination of the cause of the failure and determination of the damaged part;
- Wait for self-correction caused by organizational reasons;
- Wait for a repair associated with obtaining a corresponding spare part, etc.;
- Perform custom repairs;
- Functional inspection (check) and possible additional repair, etc. [105].

Instead of the term repair, the broader term restoration of operability, abbreviated as restoration, is sometimes used. Repairability indicators are also of two types, but different designations are used for them, e.g., the following:

- Distribution function $G(t)$ = probability of recovery in the interval <0, $t$>, density:
- Intensity of repairs (renewals) $\mu(t)$;
- Probability density of repairs (recovery) $g(t)$ [106];
- Mean time of repair (restoration of operability) (*MTTR*, mean time to recovery, or *MRT*, mean repair time);

- *p*-quantile of repair time $t_{po}$, but more often $(1 - p)$—quantile or alpha—critical value $t_{\alpha,0}$ also called guaranteed repair (recovery) time, defined explicitly as:

$$G(t_{\alpha,0}) = 1 - \alpha, \tag{19}$$

For the description of logistical, administrative, and technical delays (understood as random variables), type II indicators are most often used: mean values and *p*-quantiles. The terms used in the description of repairability are listed in Table 8.

**Table 8.** Selected terms used in the description of repairability.

| The Name of the Indicator | Designation |
|---|---|
| Repair Rate | $\mu$ |
| Mean repair time | *MRT* |
| Repair probability | $G(t)_0$ |
| Repair probability density | $g(t)_0$ |

Mean repair time (*MRT*): This belongs to the most suitable and very often used indicators of repairability. For the assumption that the intensity of repairs is constant, i.e., applying $\mu(t) = \mu$, is given by the relation:

$$\overline{t_0} = \frac{1}{\mu}, \tag{20}$$

where the following is defined:

$\mu$—the intensity of repairs (renewals).

The longer the average repair time, the worse repairability the product has and vice versa. In order to determine the time to eliminate individual faults, it is necessary to clearly determine whether only the own repair times or the total repair time, including downtime, will be considered [107].

Probability of repair $G(t)_0$: This expresses the probability of completing the repair of the product by a given time after the start of the repair, and the following applies:

$$G(t)_0 = \frac{\Delta n_0}{n_0}, \tag{21}$$

where the following are defined:

$\Delta n_0$—the number of products repaired in the interval from 0 to *t*;
$n_0$—the total number of violated products at the beginning of the monitored interval.

Repair probability density $g(t)_0$: This expresses the probability of completing the repair in an infinitesimally small time unit after a given moment, and the following applies:

$$g(t)_0 = \frac{n_0(t \to 0)}{n_0}, \tag{22}$$

where the following is defined:

$n_0$ $(t \to 0)$—the number of products repaired in the interval $t \to 0$ after time *t*.

Repair intensity $\mu(t)_0$: It is the probability of completing the repair in an infinitesimally small time unit after a given moment. Provided that the repair has not been completed by this time, the following applies:

$$\mu(t)_0 = \frac{\Delta n_0(\Delta t \to 0)}{n_0(t)}, \tag{23}$$

where the following are defined:

$\Delta n_0(\Delta t \to 0)$—the number of products repaired in the interval $\Delta t \to 0$ after time $t$;
$n_0(t)$—the number of products under repair at time $t$.

Coefficient of repairs urgency $K_{NO}$

It is given by the ratio of the number of complete failures to the total number of failures for the monitored period of operation, and the following applies:

$$K_{NO} = \frac{n_U}{n}, \tag{24}$$

where the following are defined:

$n_U$—the number of complete failures during the monitored period of operation;
$n$—the total number of failures during the monitored period of operation.

*4.8. Diagnosability*

Diagnosability is a property of the product that allows data to be obtained about its technical condition and for the necessary conclusions to be drawn from them. A comprehensive set of diagnostic actions is referred to as a diagnostic technique check, while the sequence of these actions is called a diagnostic procedure. The result of the diagnostic examination is a diagnosis, i.e., expressing a conclusion about the technical condition of the investigated object and its parts. Based on the diagnosis, a prognosis is given, i.e., a prediction of trouble-free operation for a certain time, usually until the next scheduled inspection [108].

The position of diagnostics in the technology care system can be characterized by its three basic tasks:

- Investigate the causes of obvious malfunctions that appeared suddenly and could not be predicted, resulting in a total or partial failure of the equipment;
- Check whether there is no hidden fault in the technology that could cause other dependent faults. In this case, the diagnostics have the characteristics of a regular check, aimed at detecting, for example, the tightness of joints, tightening of joints, and quality of setting of machine elements, as well as check the operation of measuring and control devices, etc.;
- In the case of gradual failures caused primarily by wear, diagnostics have an important role in monitoring the progress of wear, evaluating the technical condition, predicting the next probable trouble-free operation time, or determining the time and extent of any repair [109].

Technical diagnostics belong to the technology care system, and their procedures can be included in three groups:

- Tasks that must always be performed regularly after a certain period of operation, regardless of the degree of wear (e.g., technical maintenance): These are mainly maintenance (adjustment, lubrication, inspection) or diagnostic procedures that are related to the safe and economical operation of the equipment (e.g., regular checking of brake efficiency, engine performance check, etc.);
- Actions in which the need to perform is manifested during the diagnostic examination: These are primarily those actions for which the need results from the performance of a certain basic operation from the first (previous) group, and which fulfil the function of the so-called summary indicators of the technical condition. Their specific value is determined in each case, and it is assessed whether it is in a normal state or whether the aggregate indicator has exceeded the limit value and the cause must be sought. For example, if the engine performance is in normal condition and the engine does not smoke, there is no need to check, for instance, the injection pump, injectors, tightness of the combustion chamber and others. This procedure avoids unnecessary disassembly and assembly operations [110];
- Tasks that are performed irregularly and whose purpose it is to check the wear of technical elements where this wear is not reflected in operational economic parameters,

for example, checking the wear of crankshaft bearings: These actions are usually not performed with new technology.

- The following principles should be considered during the diagnostic procedure:
- The costs of diagnostics, the accuracy of the data obtained, and the conclusions expressed affect the effectiveness of the introduction of diagnostics. If diagnostic costs were high and data accuracy low, diagnostics would be dropped from the tech care system;
- Diagnostics should be performed with minimal disassembly of the complex product into its components, as this will shorten the diagnostic procedure, downtime of the equipment, etc., and unnecessary disassembly that worsens the quality of established connections will also be excluded;
- When creating your own procedure, it is usually expedient to first perform a diagnostic measurement, evaluating overall larger machine units, i.e., to find out first the so-called summary indicators of the technical condition, and only after determining the unit with a malfunction, identify which element is causing it. By applying this principle, the diagnostic procedure can be shortened, and thus cheaper, and unnecessary measurement can be avoided for many elements; as a rule, for summary indicators, it is sufficient to measure one or only a few values from a complex characteristic.

The level of diagnosability in the construction, production, and operation of technology allows the indicators of diagnosability to be checked [111]. A specific assessment is carried out during tests or observations of a certain set of equipment in operation.

Economic indicators can be used to assess diagnosability. These include unit time $c_d$, work $q_d$, and unit costs $u_d$ for performing technical diagnostics:

$$c_d = \frac{1}{n}\sum_1^n \frac{C_{TDi_i} + C_{TDii_i}}{T_i}\left[h \cdot T^{-1}\right], \tag{25}$$

$$q_d = \frac{1}{n}\sum_1^n \frac{Q_{TDi_i} + Q_{TDii_i}}{T_i}\left[h \cdot T^{-1}\right], \tag{26}$$

$$u_d = \frac{1}{n}\sum_1^n \frac{U_{TDi_i} + U_{TDii_i}}{T_i}\left[e \cdot T^{-1}\right], \tag{27}$$

where the following are defined:

$C_{TDi_i}$, $C_{TDii_i}$, $Q_{TDi_i}$—the total time consumption, work, and costs for performing technical diagnostics I and II $i$-th levels of technology;
$T_i$—the time of operation of the $i$-th technique until the end of the observation;
$n$—the number of observed techniques.

In some cases, diagnosability can also be assessed according to a technical indicator, the coefficient of the diagnosis structure $K_d$, which expresses the relationship between the so-called active (main) and passive (supplementary) part of diagnostics.

The actual measurement is considered the active part of the diagnosis, and the preparatory work the passive part.

The diagnostic structure coefficient $K_d$ is expressed by the relation:

$$K_d = \frac{1}{n}\sum_1^n \frac{t_{adi}}{t_{adi} + t_{pdi}}, \tag{28}$$

where the following are defined:

$t_{adi}$—the active part of the considered diagnostics of the $i$-th technique;
$t_{pdi}$—the passive part of the considered diagnostics of the $i$-th technique.

*4.9. Storability*

Storability is an important part of operational reliability for components, subgroups, groups, technology, machines, and devices. It is especially important for equipment that

works in difficult operating conditions and there are few suitable storage, garage, and parking spaces [112]. Since the equipment is exposed to various weather effects, the consumption of parts is constantly increasing, and the costs of ensuring its operability also increase. Therefore, this partial reliability property should be given extra attention.

Storability is the property (ability) of the product to maintain a fault-free condition during storage and transportation. Numerically, storability is expressed by the probability that the product will remain in a fault-free condition for the specified time of storage and transportation.

Dependability is a state in which the product fulfils or is able to fulfil the required function at a given moment.

The probability of product storability (*PS*) is calculated according to the relationship:

$$P_S = \frac{n - n_1}{n},$$ (29)

where the following are defined:

$n$—the scope of the monitored set of technology;
$n_1$—the amount of equipment on which a malfunction occurred during storage or transport.

The storability can also be expressed using unit time $t$, work $q$, and costs $u$ for storage (most often per year):

$$t = \frac{1}{n}\sum\nolimits_1^n \frac{C}{T_i} \left[h \cdot T^{-1}\right],$$ (30)

$$q = \frac{1}{n}\sum\nolimits_1^n \frac{Q}{T_i} \left[h \cdot T^{-1}\right],$$ (31)

$$u = \frac{1}{n}\sum\nolimits_1^n \frac{U}{T_i} \left[Sk \cdot T^{-1}\right]$$ (32)

where the following are defined:

$t$, $q$, $u$—the total time consumption, work, and costs associated with maintaining a fault-free state of technology during storage or transportation of the $i$-th machine;
$T_i$—a general symbol that can express the performance of the technology (in units of operation time or usage time in years, months, etc.).

By product storage, we mean the appropriate location of the product on a reserved area or in reserved spaces during non-operational periods. The special names "garaging and parking" are used for motor vehicles [113].

Storability is an important feature in the conservation and storage of technology. Damages that occur during conservation and storage of technology are caused by the following factors:

- Undesirable atmospheric influences that cause corrosion;
- Poorly executed preservation;
- Inappropriate location (in open spaces).

Equipment stored for a long time must be protected against corrosion and weather effects (rain, snow, sun, etc.), the effect of which intensifies the course of corrosion. Storing technology in covered spaces (garages, sheds) will significantly limit the effects of weather, but it will not prevent corrosion [114].

During long-term storage (parking, garage), the following basic principles must be observed:

- Clean the technique from soil residues and processed materials;
- Conserve metal surfaces;
- Repair coatings;
- Prevent tires from deteriorating;
- Appropriately store ropes, pulleys, and equipment, and clean and preserve chains;
- On machines stored outdoors, cover all non-drainage areas where water could enter.

## 5. Discussion

### 5.1. A Case Study of a Discussion on Dependability

In the following part of the study overview, the authors present a case study of the calculation principles applicability of observed quantity of reliability, i.e., dependability. In the analysis of reliability, the authors will use modeling and assimilation of the quantity reliability by using the mathematical apparatus of defined mathematical relationships.

At the present time, when relevant information on partial reliability data is available, there is an increasingly used approach to monitor the aspects of operation dependability. To evaluate dependability indicators, the authors of the presented study provide a specific example and a case study of dependability analysis using the RBD method.

Reliability modeling using RBD is therefore successfully applied to the solution for complex systems. We express the creation of a simulation model of reliability probability analysis with the following conceptual model:

- We characterize a series–parallel system by the number of subsystems in the series and by the number of subsystems of elements arranged in parallel;
- We define each subsystem by the number of elements, and for each element, we determine the value of the probability of fault-free operation;
- In the state vector of the system $XMk\varphi(t) = (X1(t), X2(t), \ldots, Xk(t))$ and its elements $Xi(t)$, the system operation time t is a constant, deterministically determined quantity. We define a simulation experiment by the number of realizations representing constant time intervals between failures;
- System elements are characterized by two basic states:

  (1)   The fault-free operational state of the PA;
  (2)   The fault condition PB.

The course and results of the experiments are presented in Figure 8.

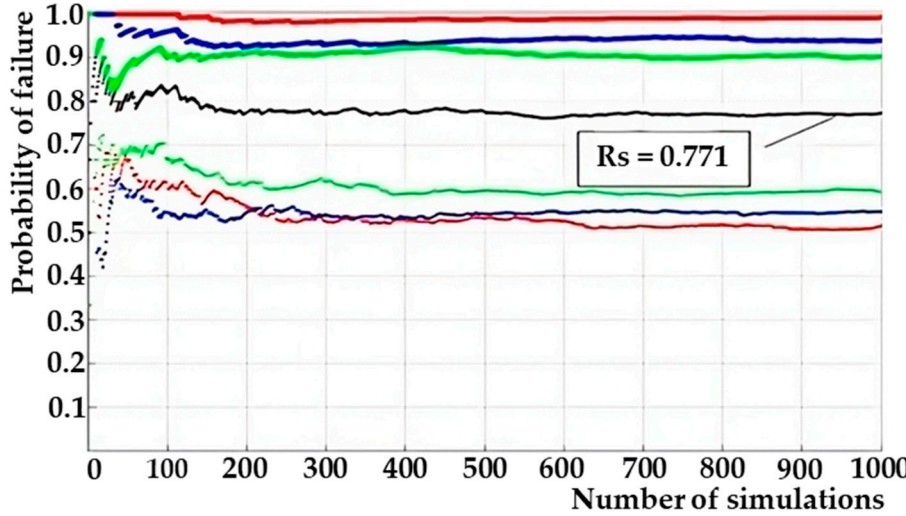

**Figure 8.** Input data, course, and results of simulation experiments with a dependability simulation model.

If the generated value exceeds the determined value of the probability of fault-free operation, the element malfunctions and vice versa. The graphical evaluation of the reliability indicators is shown in Figure 9 and the final simulation results are shown in Figure 10.

With a parallel subsystem, one functional element is enough for the subsystem to be functional; with a serial subsystem, all elements must be functional:

- The system is functional if all subsystems are functional;
- Let us denote the number of phenomena that represent the operation dependability of an element or system by n;

- If we simulate the failure of an element or system N times, the resulting probability of failure-free operation is determined by the relation R = n/N;
- The program collects output characteristics about the elements and the system.

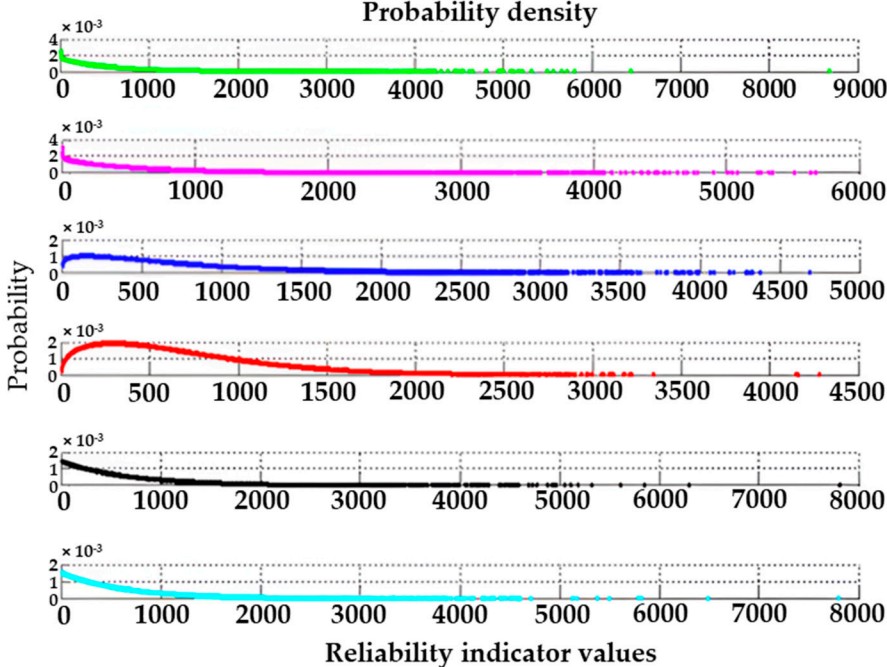

**Figure 9.** Graphical evaluation of dependability indicators.

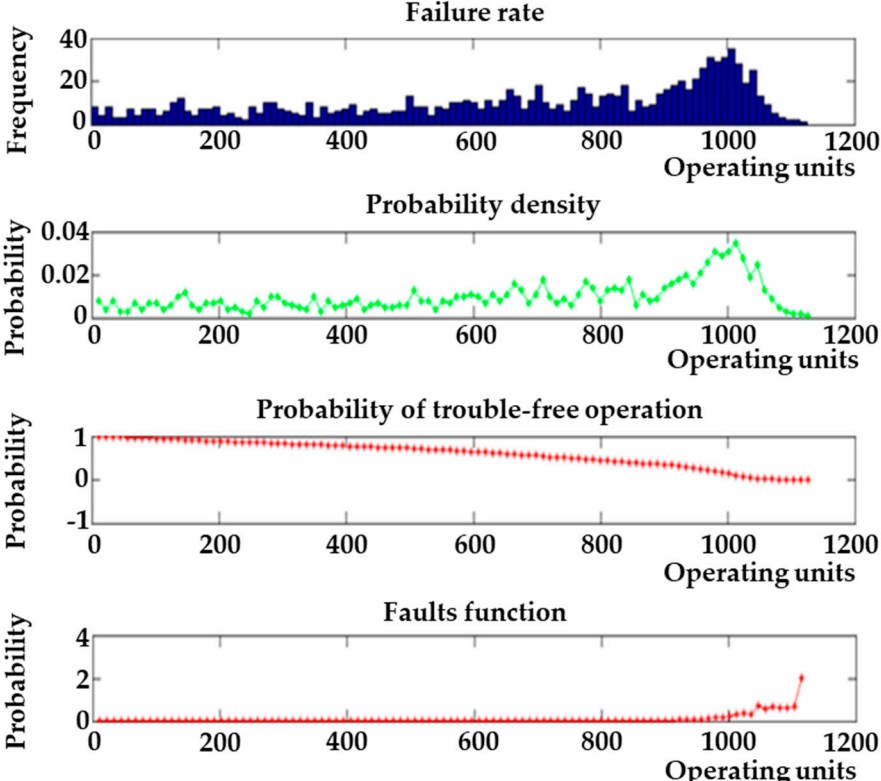

**Figure 10.** The resulting summarization of the results.

### 5.2. Case Study Discussion on Availability

In the next part of the study review, the authors present a case study of the applicability of calculation principles to the observed quantity of reliability, i.e., availability. In the analysis of availability, the authors will use modeling and assimilation of the quantity availability by using the mathematical apparatus of defined mathematical relationships. The construction of a discrete simulation model with a variable time step and the implementation of a simulation experiment for availability services require the following activities:

- Substitution of initial conditions and specification of variable values in the initial simulation time TIME = 0, and substitution of the simulation length TEND. Element state L(i) = 0, system state S = 0;
- Generation of intervals of system individual element failure from the probability distribution of times between failures x(i) (i = 1,2,...N);
- Sequencing the occurrence of faults and selecting the first event by searching for the minimum of x(i) values for i = 1, 2,..., N;
- Changing the state of element L(i) to L(j) = 1, the element is in failure, and maintenance is being carried out. Changing the status of the system S to S = 1, the system is out of order;
- Shifting the time axis by the interval of the first failure TIME = TIME + x(i);
- Generating the length of the implementation of the element maintenance operation from the probability distribution of the maintenance time y(i) (i = 1,2,..., K);
- Shifting the timeline by the maintenance period TIME = TIME + y(i). Changing the state of the element L(i) to L(j) = 0, the element is operational. Changing the status of the system S to S = 0, the system is not functional;
- Generating a new interval x(i) of failure of element i, which has been returned to serviceable condition;
- Element and system readiness calculations;
- Testing the condition of ending the simulation run, if the simulated time value reaches the pre-selected TEND value; otherwise, repeat points 3–10;
- Collecting and processing data on input and output variables using statistical methods;
- Outputting the results to the display and printer. Completion of the simulation experiment;
- The simulation model makes it possible to monitor the dynamics of changes in the state of operation and maintenance of elements and the system. Maintenance intervals are enclosed by rectangles. The course of the simulation length experiment is shown in Figure 11.

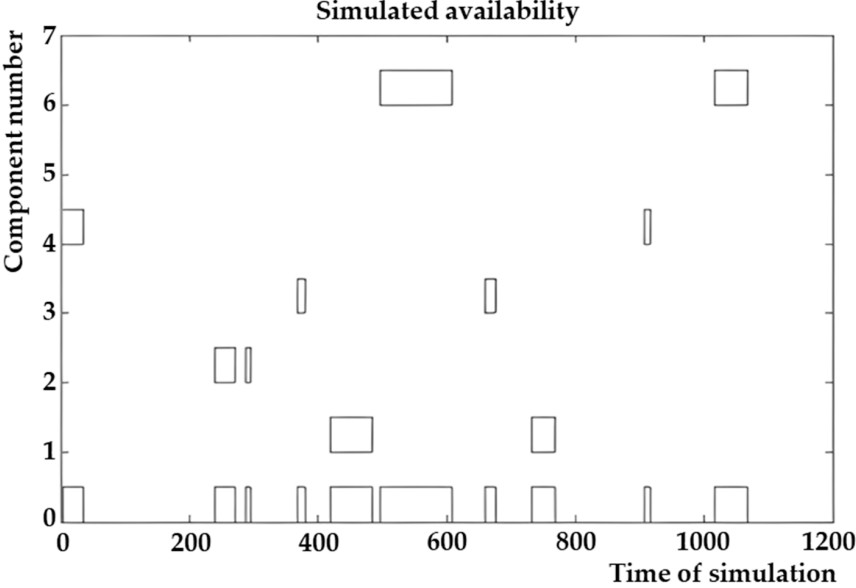

**Figure 11.** The course of the experiment with a simulation length of 267 h, with the marking of the maintenance periods of the elements and the system.

In the case of experiments with a longer simulation time, intervals with a small informative value are shown, so we evaluate the following graphs. The results of the system simulation experiment with a simulation time of 5000 h are shown in Figure 12.

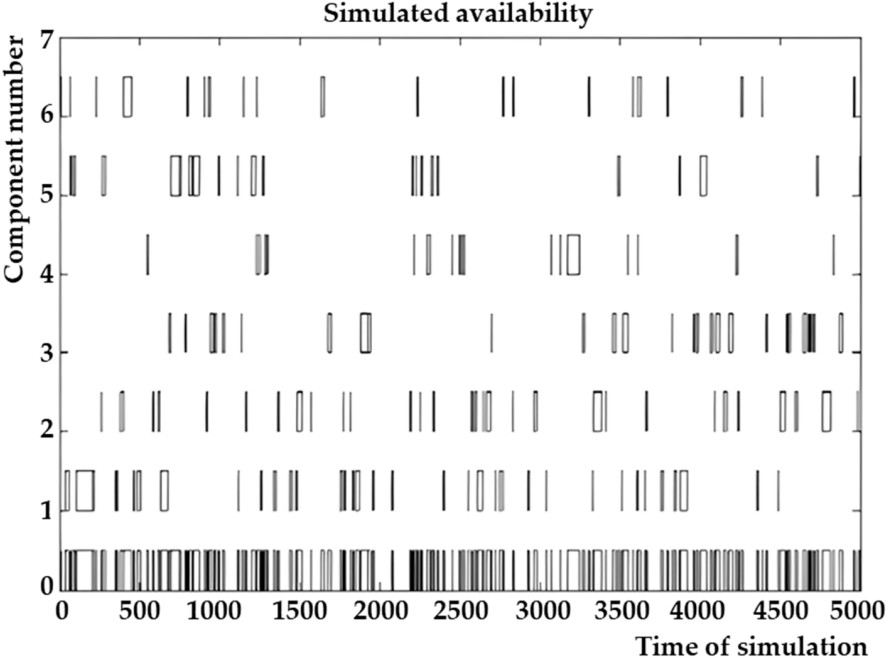

**Figure 12.** Outputs of the system simulation experiment with a simulation time of 5000 h.

Graphically, the value of the asymptotic availability factor can be observed as a function of time (Figure 13).

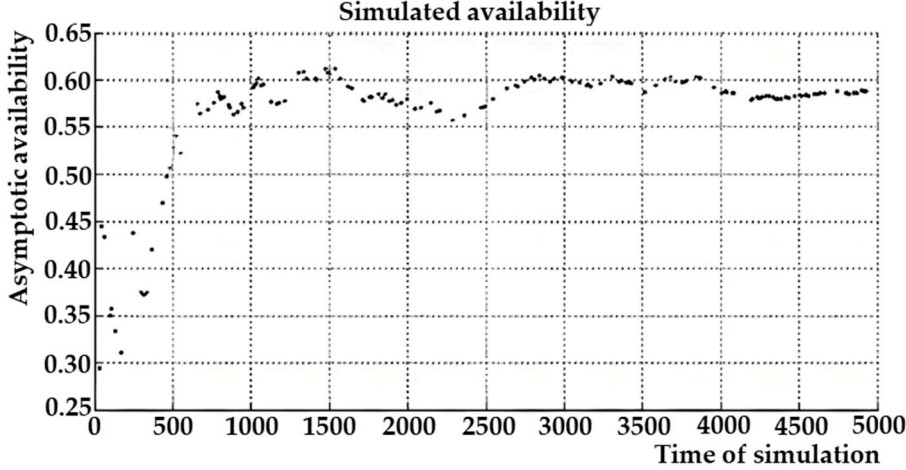

**Figure 13.** Increase in and stabilization of the asymptotic availability factor.

The rise in the value of the asymptotic availability coefficient is interesting. Contrary to claims, the time to reach a steady state is relatively long. Experiments give a certain dispersion of results with the same input values, so it is necessary to carry out a larger number of simulations. Simulation experiments are performed for a series mechanical system with six elements. Mean time between failures and mean maintenance time are exponential probability distributions.

### 5.3. Case Study Discussion on Durability

We used the methods of the interference theory of reliability (stress–strength interference method) in the modeling of lifetime values. This approach is based on the assumption

that malfunctions or faulty functions occur when the resistance limit of the object, i.e., the ability to withstand stress, is exceeded, and thus, a failure occurs. The element of the technical system to fulfill its function must be sufficiently resistant to the load that can help everyone. This is an approach based on the deliberate oversizing of the object with the expression of the safety factor against failures (SF—safety factor) or safety margin (SM—safety margin).

If the stress L exceeds the resistance value of the given structural design element, failures will occur.

The interference theory of reliability is based on the analysis of the regularities and properties of two random variables that characterize the elementary properties of dependability and lifetime. Interference reliability theory offers reliability prediction in new product design because it can simulate the various loads and stresses that are applied to the product during its life cycle. The distribution of random variables is as follows:

- The first random variable characterizes the operating mode and the resulting operating stress L (load stress). Operating stress is caused by the sum of the external stress and the conditions of the selected modes of use.
- The second random variable quantifies the strength S (carrying capacity). Strength to load S (strength) is the ability to withstand physical loads, or chemical and biological loads, which, because of their action, result in changes causing element failures.

Both parameters of the model are random variables, characterized by random variables or processes. The form of their expression can be expressed by a histogram, or after statistical processing via probability distribution functions. The literature presents many models of analytical quantification of dependability interference for the cases of exponential, normal, Weibull, gamma, or log-normal distributions of load probability densities fL (L) and strength fS (S). Representation of the range of interference is shown in Figure 14.

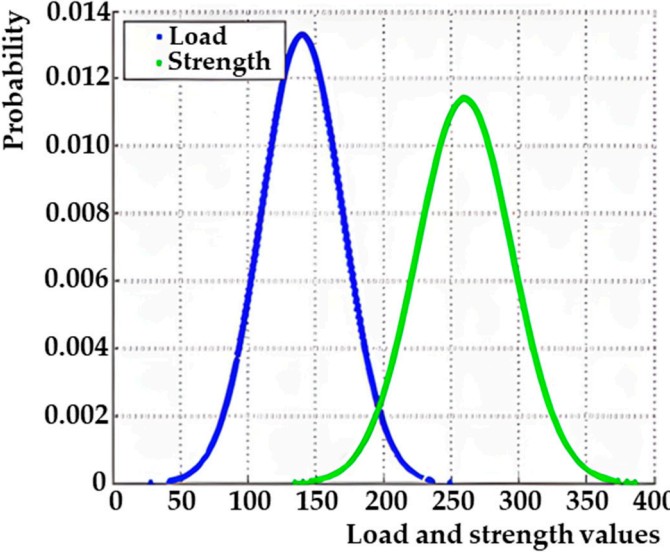

**Figure 14.** Representation of the range of interference.

The input quantities of load L and strength S have a random character obtained through experimental measurements. The result is a non-parametric distribution of the obtained data, which we can statistically process in the form of a histogram or convert to a usable parametric distribution, as illustrated in Figure 15. Both cases provide us with the possibility of generating input quantities and assessing the occurrence of decisive events for the statistical expression of failure rate or failure-freeness of elements using the interference method. The histogram representation of the random variables L and S is shown in Figure 15.

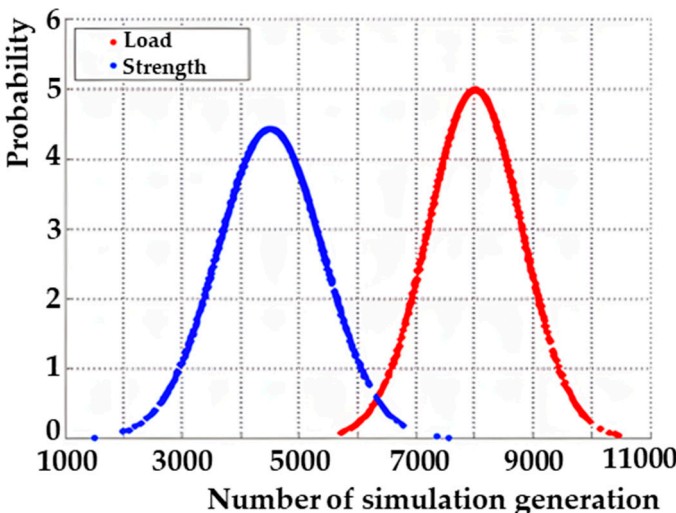

**Figure 15.** Expression of random variables L and S using histogram of relative abundance and probability distribution density.

For the range of experimental or generated values, we determine the size of the values of the distribution functions FL(S) and FS(S) for different strength values S. For the range of both functions' values, stress L and S contribute to failure. If we plot the values of L and S in the interdependence graph, the intersection represents the product of two independent phenomena. The area below the line of the graph represents the probability of the operation dependability expressed, and the area above the line of the graph represents the probability of the occurrence of a fault. The failure probability is shown in Figure 16.

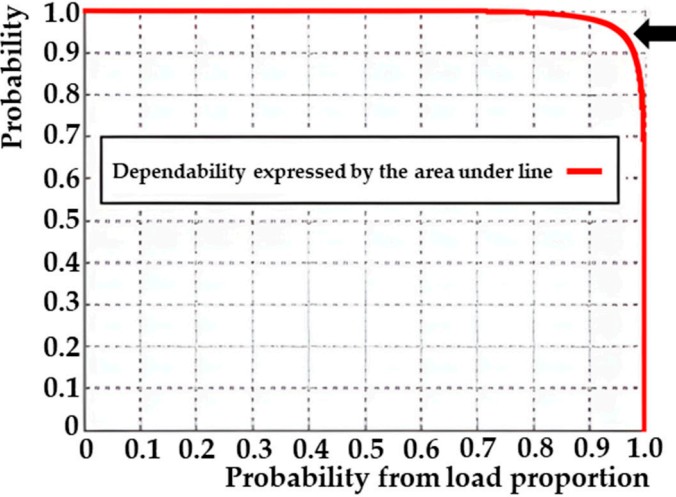

**Figure 16.** Probability of failure.

Load L is a stochastic quantity with properties. It has its distribution of the probability of occurrence at individual levels, which do not change its character (type and parameters of the distribution) with time (period of operation). The resistance of the structure to failure S with time does not change its type (law) of distribution but changes its position relative to the origin of the coordinates. A change in position occurs when the stress repeatedly exceeds a certain threshold limit Sc of the structure sensitivity (resistance). The application of the dynamic model requires the clarification of some important concepts and properties of the random variables used in the model, above all, the clarification of the stochastic nature of the quantities S and L, especially their possible change with the time of stress exposure, and further the concept of "accumulation of damage". The possibilities of

variations in how the system will react to different strength need to be verified via repeated modelling. Ongoing analyses can be seen in Figure 17.

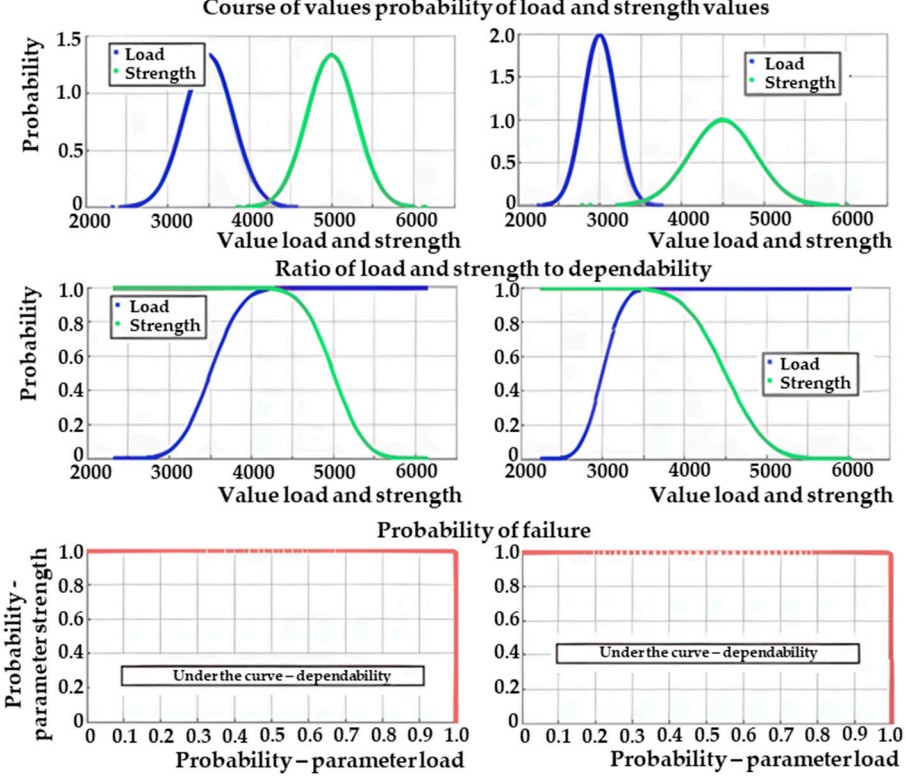

**Figure 17.** Principle of the impact of changes in the strength parameter.

## 6. Conclusions

In today's technologically advanced society, there is no longer an area of the real world in which reliability predictions are not used to achieve the necessary system security. The importance of reliability analyses and the monitoring of partial reliability properties is growing more and more, and therefore, more and more demands are placed on reliability analyses. Not only on their functionality, which they can provide to users, but mainly on their quality, modifiability (in view of changing customer requirements) and the ability to communicate with surrounding systems. In the past, the overall quality of the system was evaluated only after its creation. Today, this method is already on the decline, and a method that tries to minimize the failure rate of the final product already in the stages of system development is beginning to be promoted. Nowadays, reliability is considered a very important feature of every product. By product, we can understand this to refer to one specific product, object, service, or even an entire extensive system. Reliability parameters undoubtedly significantly affect the quality of the product on the market and its benefit. Every product has its own life cycle, in which it is affected by a considerable number of variable factors. At each stage, there are mutual interactions with the environment, the user, the operating environment, and related processes in various areas of human activity (economics, technology, production management, marketing, human resources, etc.). At each stage of the life cycle, it is necessary to monitor and predict partial reliability properties. In real life, of course, there is no sequence, but on the contrary, everything is interconnected. This process is constantly changing and dynamic. Activities in individual departments affect the functional properties and qualitative features of the product (and thus also its technical and economic success). On the other hand, the variability of the product, application environment, economic factors of the market, etc., affect processes in individual sector areas. Reliability can also be understood as a time component of quality, so reliability is a subset of quality. When preparing new systems and when evaluating

the reliability of systems in operation, it is necessary to quantify some of the reliability indicators. As mentioned in the previous chapters, we must understand reliability as a composite property composed of partial properties, but in general, when predicting reliability, we must focus on the evaluation of basic indicators. At the conclusion of the presented review, the idea can be established that for every company or field, creating the most reliable and fault-free system is an important task and challenge. Therefore, it is necessary for reliability analysts to have objective data, from which the analysis will be carried out, to be able to assess the reliability of all their products and, in the event of quality deterioration or stagnation, to be able to take appropriate steps leading to the correction of this condition throughout the entire product life cycle. Humanity's dependence on technology and innovative products is growing every day, and therefore, it is necessary to ensure that their failure rate is minimized with maximum safety in mind. Not only because of the possible loss, but especially because any failure can result in the loss of human life, which cannot be returned. If deploying a malfunctioning system can result in such a situation, it is essential that the risk of its occurrence be minimized. At the same time, it is necessary that in the event of a failure, the system should be able to return to normal operation as independently as possible.

**Author Contributions:** Conceptualization, A.B.; methodology, M.E.; software, M.E.; validation, A.B.; formal analysis, M.K. (Marcel Kohutiar); investigation, A.B.; resources, P.M.; data curation, M.K. (Michal Krbaťa); writing—original draft preparation, A.B.; writing—review and editing, P.M.; visualization, M.K. (Marcel Kohutiar); supervision, M.K. (Michal Krbaťa); project administration, M.K. (Marcel Kohutiar); funding acquisition, M.K. (Michal Krbaťa). All authors have read and agreed to the published version of the manuscript.

**Funding:** This research received no external funding.

**Data Availability Statement:** Data are available upon request to the corresponding author.

**Conflicts of Interest:** The authors declare no conflict of interest.

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
