# Peer review of "Reliability Analysis during the Life Cycle of a Technical System and the Monitoring of Reliability Properties"

_systems, doi:10.3390/systems11120556_

Round 1
Reviewer 1 Report
Comments and Suggestions for Authors
1.Please highlight the technical contributions.
2.A case study should be added to verify the proposed approach.
3.Please specify the technical system,different types have different demands.
4.Newly published related papers should be added and discussed,especially from this journal.
Comments on the Quality of English Language
Minor editing of English language required.
Reviewer 2 Report
Comments and Suggestions for Authors
The study underscores the importance of ongoing reliability analysis in technical systems to ensure safety and minimize potential failures, especially given the high stakes of technological dependence. Here is my comment:
The introduction should provide a more in-depth literature review, showcasing seminal works, recent developments, and any notable trends in reliability analysis. By doing so, the readers can ascertain the context in which this research is situated. Furthermore, pinpointing the specific gap this study addresses compared to existing literature would highlight its unique contribution and significance.
Reviewer 3 Report
Comments and Suggestions for Authors
The authors in the work titled "Reliability analysis during the life cycle of a technical system & monitoring of reliability properties" aim to highlight the importance of reliability analysis during the lifecycle of a physical asset, and identify various techniques and concepts relating to reliability analysis. The authors have tried to strengthen the work using many fundamental concepts, which is good however, the work is weak due to the following reasons.
1. In many places, authors repeat concepts, sentences, and ideas. This is not a good way of writing academic papers.
2. Authors have talked about fundamental concepts which are already widely known.
3. There are no demonstrations of fundamental knowledge usage.
4. Some concepts like reliability block diagrams are not discussed.
5. Would recommend that authors simplify sentences, which on many occasions appear to be lengthy, complicated, and lose meaning.
6. Discussion of case studies, application of reliability concepts, and references to major achievements using reliability analysis will benefit a revised version of this work.
Comments on the Quality of English LanguageIn its present form, the work is very weak.
Reviewer 4 Report
Comments and Suggestions for Authors
This manuscript offers a summary of reliability analysis methods and concepts, focusing on the examination and forecasting of reliability indicators. It establishes a mathematical foundation for individual sub-indicators and outlines the process of component/system reliability analysis, which encompasses the acquisition, examination, and organization of system-specific information for objective assessments of overall reliability levels. Despite the manuscript's coverage of reliability indicators, mathematical definitions and expressions, and the significance of reliability within a technical system's life cycle, the reviewer has questions, suggestions, and concerns. Therefore, the reviewer recommends a major revision before considering the manuscript for acceptance.
1. To begin with, a thorough revision of the manuscript is necessary to ensure the clarity of sentences. Many sentences are composed in an abstract and vague manner, making it challenging for reviewers to discern the central points of these sentences
2. In line 45, it states that “ To achieve the required level of reliability, we use reliability analyses,…” however, these reliability analyses are used for assessing the system's reliability. To attain the desired level of reliability, separate steps involving design modification, iteration, and optimization are undertaken. Hence, this statement requires additional clarification
3. In numerous instances throughout the manuscript, the term "The presented research" is used. However, since this is a review article without original research, the meaning and relevance of "the presented research" remain unclear.
4. The introduction should incorporate references to existing review articles on the reliability of systems, particularly those that concentrate on the entire lifecycle of a system. Consequently, the manuscript should explicitly state the objectives of these review articles and underscore its unique contributions and value in relation to the existing body of review literature.
5. It states that “The procedure is shown in Figure 1 [26].” However, Figure 1 is not from the reference [26]. Consequently, the rationale for including this particular reference remains unclear.
6. What does 'FTA' stand for? Please provide the full expansion of the acronym the first time it is used. Similarly, ‘IEC’
7. “Graphic processing in Figure 3” is not a full sentence.
8. The overall figure legend is notably concise and lacks sufficient information. It is crucial to understand that figure legends should be comprehensive enough to assist readers in interpreting the information presented in the figure.
9. The manuscript includes mathematical expressions that pertain to various criteria, including reliability, storability, diagnosability, repairability, and more. However, the manuscript lacks information regarding the specific methods and approaches for computing these criteria. For instance, there exist various reliability analysis techniques such as FORM, SORM, active learning-based approaches, Monte Carlo simulation, and subset simulation, among others. Similarly, there are numerous approaches for computing diagnosability and repairability. To ensure completeness and achieve the objectives of the manuscript, it is imperative to include details about these methods and approaches.
10. The manuscript mainly centers on the presentation of reliability variables. It adopts a format that briefly outlines definitions and mathematical expressions, which, however, does not comprehensively cover each topic. A comprehensive review article should encompass research findings, reference pivotal articles, elucidate areas of consensus, highlight existing gaps in knowledge, indicate unresolved questions, and propose future research directions. Hence, it is essential to include specific examples, challenges, trials, advancements, and more, going beyond mere summarizations with limited mathematical expressions and explanations of each component, as these cannot capture the full scope of each respective field. Hence, providing insights into the critical advancements in each reliability indicator within Section 4 and presenting the state-of-the-art methods will offer readers more informative content.
11. Given the potentially broad nature of a technical system's life cycle and the strong dependence of developed research methods and applications on problem types and specific systems within a subject field, it would be beneficial to establish a defined scope for technical systems (mechanical, structural, material, infrastructure system, etc). This way, all reliability indicators can be reconfigured with a specific focus on these defined technical systems
Comments on the Quality of English LanguageSome degree of English revision is recommended.
Round 2
Reviewer 1 Report
Comments and Suggestions for Authors
The paper is revised carefully,the acceptance is suggested.
Comments on the Quality of English LanguageThe paper is revised carefully,the acceptance is suggested.
Reviewer 4 Report
Comments and Suggestions for Authors
The reviewer's concerns and suggestions have been incorporated into the revised manuscript.
Comments on the Quality of English Languageits okay